# Soybean reduced internode 1 determines internode length and improves grain yield at dense planting

Shichen Li[1,2,5], Zhihui Sun[1,5], Qing Sang[1,5], Chao Qin[3,5], Lingping Kong [1], Xin Huang[1], Huan Liu[1], Tong Su[1], Haiyang Li[1], Milan He[1], Chao Fang[1], Lingshuang Wang[1], Shuangrong Liu[1], Bin Liu [3] ✉, Baohui Liu [1,2] ✉, Xiangdong Fu [4] ✉, Fanjiang Kong [1,2] ✉ & Sijia Lu [1] ✉

Major cereal crops have benefitted from Green Revolution traits such as shorter and more compact plants that permit high-density planting, but soybean has remained relatively overlooked. To balance ideal soybean yield with plant height under dense planting, shortening of internodes without reducing the number of nodes and pods is desired. Here, we characterized a short-internode soybean mutant, *reduced internode 1* (*rin1*). Partial loss of *SUP-PRESSOR OF PHYA 105 3a* (*SPA3a*) underlies *rin1*. RIN1 physically interacts with two homologs of ELONGATED HYPOCOTYL 5 (HY5), STF1 and STF2, to promote their degradation. RIN1 regulates gibberellin metabolism to control internode development through a STF1/STF2–*GA2ox7* regulatory module. In field trials, *rin1* significantly enhances grain yield under high-density planting conditions comparing to its wild type of elite cultivar. *rin1* mutants therefore could serve as valuable resources for improving grain yield under high-density cultivation and in soybean–maize intercropping systems.

In 2021 alone, almost 371.7 million tonnes of soybean [*Glycine max* (L.) Merr.] were produced from an estimated 120.5 million ha of cultivated land worldwide (https://www.fao.org/faostat/en/#home). Soybean not only provides a large amount of edible oil and vegetable protein for humans, but is also the main source of protein for animal feed[1,2]. However, grain yield of soybean at the population level is far behind certain staple crops, such as rice, wheat and maize. These cereals have benefitted from semi-dwarf phenotypes, which reduce the risk of lodging and thus enabling rice and wheat to be grown under high-density planting conditions that can lift yields[3,4]. However, unlike rice and wheat, plant height in soybean is determined by node number and

internode length, and a decrease in plant height is often because of fewer nodes. Soybean is a pod crop, and because pods adhere to the nodes, a decrease in node number leads to a decrease in pod number, and consequently reduces yield[5]. Therefore, obtaining shorter plants that retain their node number is a potential way to lift soybean yield[6–8]. This could potentially be realized with shorter internodes that allow plants to adapt to high-density planting.

Reduced plant height and lodging tolerance are critical for dense planting[9]. Gibberellins (GA) are a class of plant-growth hormones and well-known determinants of plant height. In rice, the *semi-dwarf1* (*sd1*) allele, which encodes *GA20-oxidase 2* (*GA20ox2*), confers the

[1]Guangdong Key Laboratory of Plant Adaptation and Molecular Design, Guangzhou Key Laboratory of Crop Gene Editing, Innovative Center of Molecular Genetics and Evolution, School of Life Sciences, Guangzhou University, Guangzhou 510006, China. [2]The Innovative Academy of Seed Design, Key Laboratory of Soybean Molecular Design Breeding, Northeast Institute of Geography and Agroecology, Chinese Academy of Sciences, Harbin 150081, China. [3]The National Key Facility for Crop Gene Resources and Genetic Improvement (NFCRI), Institute of Crop Science, Chinese Academy of Agricultural Sciences, Beijing 100081, China. [4]State Key Laboratory of Plant Cell and Chromosome Engineering, Institute of Genetics and Developmental Biology, Innovation Academy for Seed Design, Chinese Academy of Sciences, Beijing 100101, China. [5]These authors contributed equally: Shichen Li, Zhihui Sun, Qing Sang, Chao Qin. ✉e-mail: liubin05@caas.cn; liubh@gzhu.edu.cn; xdfu@genetics.ac.cn; kongfj@gzhu.edu.cn; lusijia@gzhu.edu.cn

semi-dwarf phenotype[10,11]. The introduction of *Reduced height* (*Rht*)-*B1b* and *Rht-D1b* mutant alleles, which encode N-terminally truncated DELLA proteins, function as a suppressor of GA signaling to regulate plant height in wheat[3,12]. Both *sd1* and *Rht* are so-called 'Green Revolution genes' that dramatically reduce plant height to suit high density planting and use nitrogen efficiently, thus doubling the grain yield of rice and wheat[10–12]. In soybean, several genes have been reported to regulate plant height through the gibberellin pathway, but few studies have focused on internode length[13–15]. The CRY1–STF (HY5)–gibberellin regulatory module regulates soybean height and internode length through gibberellin metabolism at high growth densities, in response to low levels of blue light[16]. Under normal light conditions, blue light activates CRY1 proteins, which in turn promote the accumulation of bZIP transcription factors STF1 and STF2 that share homology with Arabidopsis ELONGATED HYPOCOTYL 5 (HY5). These STFs then directly activate *GA2ox* expression, leading to a decrease in $GA_1$ levels, to ultimately suppress the elongation of soybean internodes. Conversely, when blue light is reduced under high planting density, CRY1 proteins are deactivated, resulting in a down-regulation of *STF1*, *STF2* and *GA2ox* expression, to ultimately promote soybean internode elongation[16].

In Arabidopsis, the E3 ubiquitin ligase CONSTITUTIVELY PHOTOMORPHOGENIC 1 (COP1) forms a tetrameric complex with members of the SUPPRESSOR OF PHYA 105 (SPA) family and interacts with HY5, resulting in the ubiquitination of HY5 and its degradation that eventually leads to suppression of photomorphogenesis in darkness[17–21]. Four *SPA* genes are present in the Arabidopsis genome, namely *SPA1* through *SPA4*. All SPA proteins have a WD-repeat domain, a coiled-coil domain, and a kinase-like region[22]. Of these domains, the WD-repeat domain shows the highest sequence similarity among members[23]. The activity of both COP1 and SPA proteins requires WD-repeat domains, indicating that WD-repeat domains are necessary for the function of the COP1–SPA complex[24–26]. In addition, SPA1 can also independently interact with HY5 and reduce HY5 protein levels in the light[17,27]. However, whether SPA proteins contribute to plant height and internode length in soybean remains unknown.

Here, we identified a soybean mutant with shorter internode lengths and plant height that we named *reduced internode 1* (*rin1*). Genetic and molecular characterization of *rin1* suggest that it encodes a partial loss-of-function homolog of Arabidopsis *SPA3*. RIN1 interacts with STF transcription factors to facilitate their degradation by RIN1, to regulate downstream *GA2ox* gibberellin-oxidase genes that control internode length. Interestingly, we found that the *rin1* allele reduces internode length without reducing per-plant yield, but increases population yield/plot yield under high-density conditions compared to wild type. Our findings uncover the genetic basis of *RIN1* for control of internode length and provide a promising allele for molecular breeding to improve yield for soybean high-density planting and potential use in soybean–maize intercropping.

## Results

### Identification and characterization of *rin1* mutant

Cultivated soybean originated in the middle latitudes of China, but Heilongjiang province in northeast China is becoming a major soybean production area in China in that account for 40% of total production. Heinong 35 (HN35) is an elite cultivar grown widely in Heilongjiang province because of its relatively high yield and high protein content. To identify additional genes contributing to soybean plant architecture, we generated a HN35 mutant library by γ-ray irradiation. After evaluation and screening, we found a dwarf mutant with extremely short internodes (Supplementary Fig. 1a–d) and compact plant architecture. Such an architecture may have the potential to increase soybean yield under high-density planting. We named this mutant as *reduced internode 1* (*rin1*). Interestingly, the per-plant yield of *rin1* mutant is higher than that of wild-type HN35 (Supplementary

Fig. 1e–h). This indicates the *rin1* mutant is a potential resource for breeding both semi-dwarf and high-yield cultivars.

To identify the candidate gene underlying *rin1*, we crossed *rin1* and Heihe 43 (HH43), another elite cultivar grown in Heilongjiang province, and generated a $F_2$ segregating population, and named it HNH population. Compared with *rin1*, HH43 has taller plants, longer internode lengths, fewer nodes and a higher hundred-grain weight (Supplementary Fig. 2a–d, h). However, the total grain weight and the number of grains per plant were less than those of *rin1* (Supplementary Fig. 2e–g). Based on the Genotyping-by-Sequencing (GBS) data, internode length and plant-height data of the $F_2$ population, we mapped the *rin1* locus to chromosome 12 (Fig. 1a, b; Supplementary Tables 1, 2). $F_{2:3}$ genetic testing further confirmed the $F_2$ mapping result in which *rin1* is located on chromosome 12 (Fig. 1a, b; Supplementary Tables 1, 2).

### Positional cloning of *rin1*

To clone the candidate gene, we then generated a large residual heterozygous inbred population from the HNH population that only segregates at the *rin1* locus. Combining the phenotype and genotype of the recombinants, we fine-mapped the *rin1* locus to a 76.31 kb region, in which seven genes are annotated in the Williams 82 (Wm82) reference genome (Fig. 1c, d; Supplementary Table 3). Among these, only one gene *Glyma.12G224600.1*, a homolog of Arabidopsis *SPA3* (named *SPA3a* in soybean), bears a mutation in the coding region between the parents *rin1* and HH43 (Fig. 1e; Supplementary Table 3). Compared with HH43, the *rin1* mutant harbors two non-synonymous variations at the coding sequence (CDS) positions of 223 and 2,521 (Fig. 1e). The single-nucleotide polymorphism (SNP) at position 223 causes a conversion of the 75th amino acid from glycine to serine (G–S) in the first exon, and the SNP at position 2521 generates a premature stop codon in the sixth exon of *SPA3a*, causing truncated SPA3a protein (Fig. 1e, Supplementary Fig. 3). In addition to the SNP at position 223, another non-synonymous variation at position 2,654 is found in two normal-internode cultivars HH43 and HN35 (Fig. 1e), suggesting that both SNP 223 and 2654 are nonsense mutations with roles in internode development. We therefore considered that *SPA3a* is the candidate gene for *RIN1* in which the SNP at position 2521 with a premature stop codon is the causative mutation of *rin1*. We thereafter named *RIN1* in Wm82 and HH43 as *RIN1^Wm82* and *RIN1^HH43*, respectively.

To validate the function of *SPA3a* in soybean, we generated two knockout mutants of *RIN1* by CRISPR/Cas9 in the Wm82 background. To differentiate the knockout mutants from *rin1*, we named them as *rin1^CR1* and *rin1^CR2* (Supplementary Fig. 4a–c). Both *rin1^CR1* and *rin1^CR2* encode very short truncations (53 and 54 amino acids in length, respectively) lacking the bulk of kinase domain, the whole coiled-coil domain, and the whole WD40-repeat domain (Fig. 1e, Supplementary Fig. 4d–f). Due to the paleopolyploid nature of the genome, soybean encodes 10 predicted SPA homologs with high protein similarity with the Arabidopsis homologs (Supplementary Fig. 5). To exclude the possibility that *RIN1* homologs bore off-target mutations, we sequenced the 10 *SPA* genes at the mutation location of *rin1^CR*. *SPA3b* gene which contains the same target sequences as *RIN1* was not mutated in *rin1^CR* mutant (Supplementary Figs. 6, 7) and the remaining *SPA* genes in soybean were also not targeted in the *rin1^CR* mutant (Supplementary Data 1 and Supplementary Data 2). In a growth chamber with non-inductive long-day photoperiods that delay soybean flowering, *rin1^CR1* and *rin1^CR2* both had pronounced dwarfism accompanied by fewer nodes and shorter internode lengths (Fig. 2a–d), thus further confirming that *SPA3a* is the causative gene of *RIN1* that determines node number and internode length, and in turn plant height. To further support this, we developed a pair of near-isogenic lines NIL-*RIN1^HH43* (with the allele from HH43) and NIL-*rin1* from the HNH population (Supplementary Fig. 8a, b). As expected, NIL-*rin1* showed the dwarf phenotype, had fewer nodes and shorter internodes under long-day photoperiods when grown in a field in

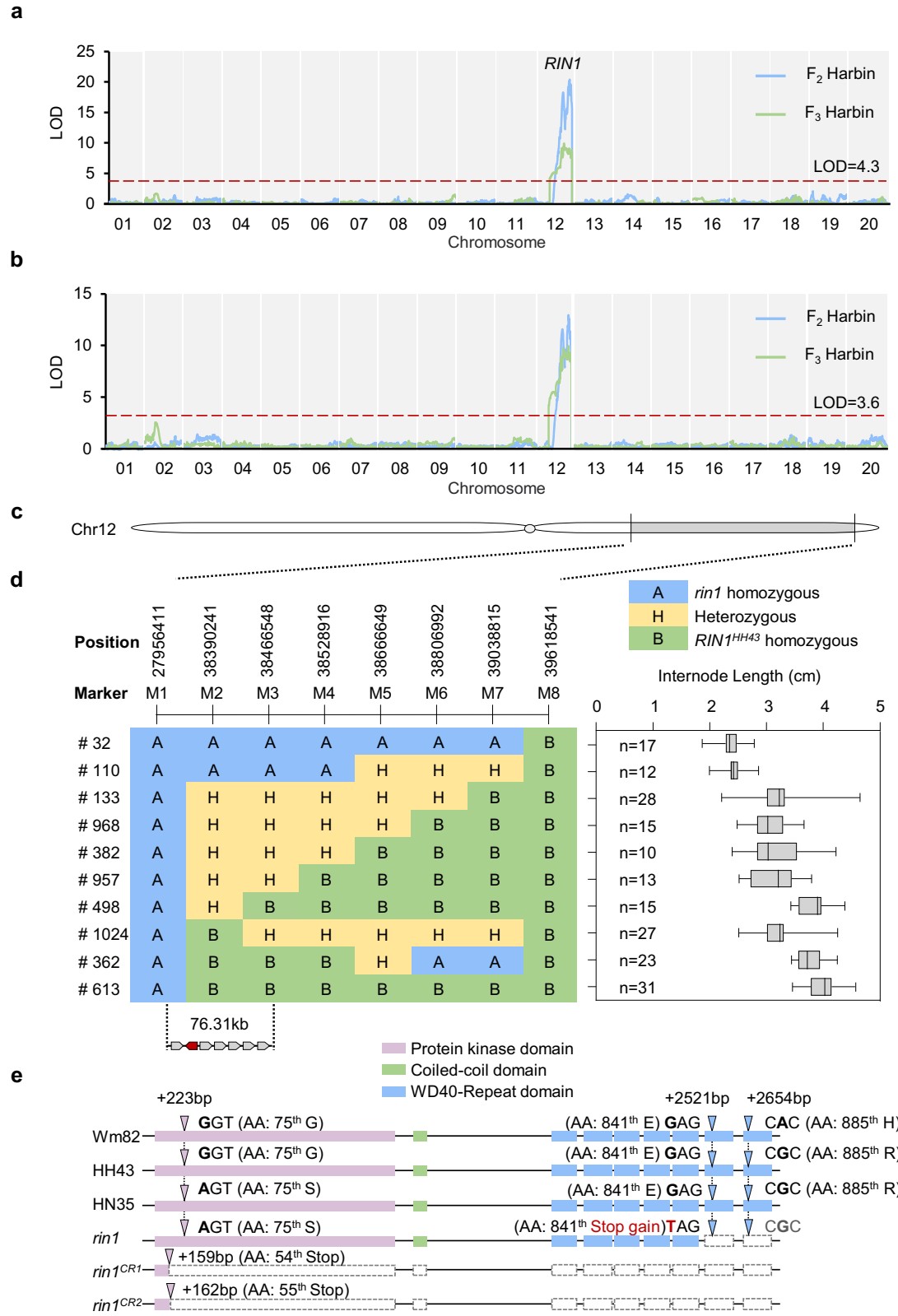

Shijiazhuang, China (Fig. 2e–h). These results collectively demonstrate that *RIN1* functions as a positive regulator of node number and internode length, which together determine soybean height.

Inspired by the role of genetic variation in germplasm contributing to crop domestication and improvement, we analyzed natural variation in *RIN1* using re-sequencing data from a panel of 1295 soybean accessions[28]. The non-synonymous SNPs at positions 223 and 2654 led us to define three haplotypes in which haplotype 1 (*RIN1*-H1) is the same as the Wm82 reference genome, *RIN1*-H2 corresponds to HH43, and *RIN1*-H3 is the HN35 haplotype (Supplementary Fig. 9a). 297 cultivated accessions (137 landraces and 160 improved cultivars) were assessed for plant performance in the field at three locations in China – Zhengzhou at 34°N, Wuhan at 31°N, and Guangzhou at 23°N. The non-synonymous substitutions at positions 223 and 2,654 have statistically

**Fig. 1 | Identification and positional cloning of *rin1*. a** Whole-chromosome scan of QTLs for plant height both in F₂ (Harbin, 2017) and F₃ (Harbin, 2018) populations. The red dotted line represents the threshold for QTL detection, LOD (F₂) = 4.3, LOD (F₃) = 3.2. **b** Whole-chromosome scan of QTLs for internode length in both F₂ and F₃ populations. The red dotted line represents the threshold for QTL detection. LOD (F₂) = 3.6, LOD (F₃) = 3.4. **c** Schematic illustration of the chromosomal location of the QTL on chromosome 12. **d** Fine-mapping of *rin1* to a 76.31 kb region. Characterization of key recombinants in the immediate vicinity of the *RIN1* locus showing recombination break points (left panel). A: Homozygous for the allele from *rin1*; B: homozygous for the allele from HH43; H, heterozygous. Segregation

of internode length is shown in boxplot format (right), wherein the interquartile region, median and range are represented by the box, bold vertical line, and horizontal line, respectively. All the plants used for phenotypic measurements in (**d**) were planted in a field in Harbin, China (45°75′N, 126°63′E) and the phenotypes were scored after maturity. n represents the number of plants of each genotype. **e** Gene structures of the *rin1* candidate gene show three allelic variations in Wm82, HH43, HN35 and *rin1*. +, coding regions (CDS). AA, Amino acid. The pink bars represent Protein kinase domain, green bars represent Coiled-coil domain and blue bars represent WD40-Repeat domain. The triangle symbol represents the base mutation position. Source data of (**d**) are provided as a Source Data file.

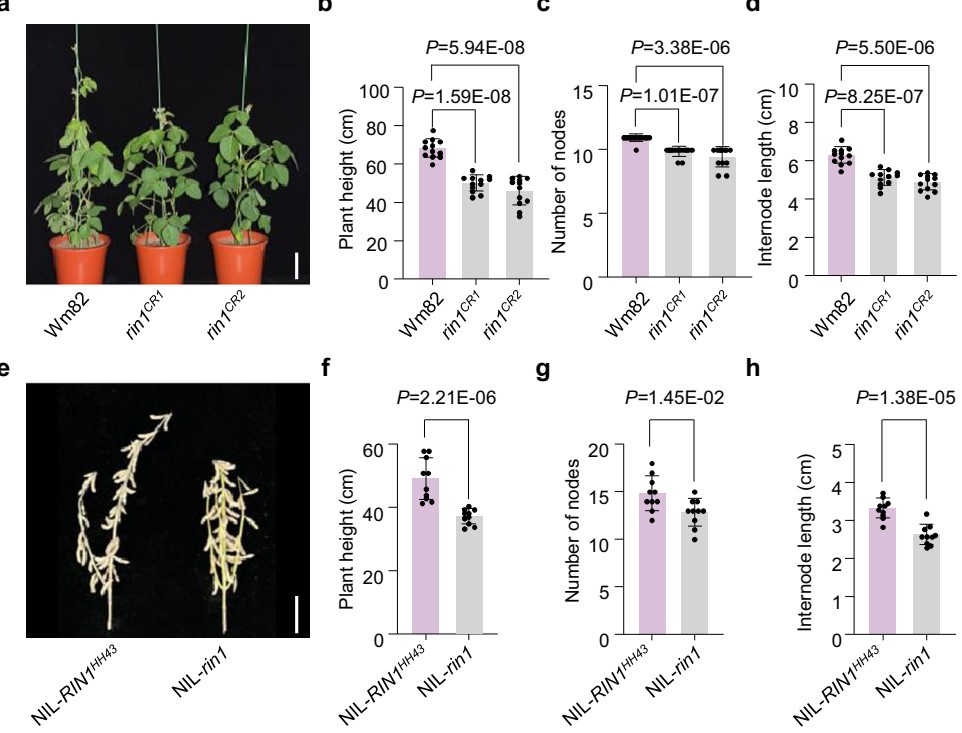

**Fig. 2 | Genetic validation of *RIN1*. a–d** Phenotypic characterization of wild-type Wm82, *rin1^CR1* and *rin1^CR2* mutants grown in long-day photoperiods. **a** Phenotype of Wm82, *rin1^CR1* and *rin1^CR2* mutants at 40 DAE. Scale bar = 10 cm. **b** Plant height (cm). **c** Number of nodes. **d** Internode length (cm). All data of (**b–d**) are means ± SEM (*n* = 12 plants). A student's *t*-test (two-sided) was used to generate the *P*-values. Plants in (**a–d**) were grown in a growth chamber under a regime of 16 h light and 8 h darkness with three plants per pot. **e–h** Phenotypic characterization of NIL-*RIN1^HH43* and NIL-*rin1* plants grown under long-day photoperiods. **e** NIL-*RIN1^HH43* and NIL-*rin1*

plants at maturity. Scale bar = 10 cm. **f** Plant height (cm). **g** Number of nodes. **h** Internode length (cm). All data of (**f–h**) are means ± SEM (*n* = 10 plants). A student's *t*-test (two-sided) was used to generate the *P*-values. Plants in (**e–h**) were grown in a field in Shijiazhuang, China (37°27′N, 113°30′E) in 2022. Each row was 2 m long, with 60 cm spacing of between rows. 30 plants were sown in each row and phenotypes were scored after maturity. Source data are provided as a Source Data file.

non-significant effects on internode length and plant height (Supplementary Fig. 9b–g), but do have statistically significant effects on flowering time under these conditions (Supplementary Fig. 9h–j). This suggests that *RIN1* may also have roles in flowering time and regional adaptation in soybean. We examined the flowering time of Wm82 and the *rin1^CR1* null mutant under long-day photoperiods in a growth chamber. The *rin1* mutation resulted in earlier flowering (Supplementary Fig. 10a, b). These results also indicate that the molecular mechanisms for flowering and internode development mediated by *RIN1* may be acting independently.

### *RIN1* is highly expressed in shoot apical meristems during the early vegetative stage

To characterize *RIN1* expression patterns and potential functional relevance, we performed RT-qPCR and in situ hybridization experiments in soybean throughout shoot apical meristem development. *RIN1* is strongly expressed in the shoot axillary meristems, trifoliate

leaves and axillary meristems (Supplementary Fig. 11a). Because the shoot apical meristem is the key organ determining plant height and internode development[29], we therefore elected to focus on this organ to more closely catalog the expression profile of *RIN1* during vegetative growth. We first confirmed the specificity of the *RIN1* probe used for in situ hybridization. Compared with wild-type Wm82, *RIN1* expression nearly cannot be detected in *rin1^CR1*, but the expression of *SPA3b* (which has the highest homology with *RIN1*) was not obviously affected either *RIN1* or *rin1^CR* lines (Supplementary Fig. 11b, c). Next, we performed in situ hybridizations in shoot apical meristems isolated from Wm82 at 10 days after emergence (DAE), 20 DAE and 30 DAE. *RIN1* is expressed in the shoot apical meristem, leaf primordia and axillary meristems (Fig. 3a). We also performed RT-qPCR in shoot apical meristems every 5 days after emergence from 5 to 40 DAE. *RIN1* is detectable at 5 DAE, peaks at 10 DAE, and then decreases gradually (Supplementary Fig. 11d), suggesting importance in the early stages of shoot-apical-meristem development. Additionally, we evaluated *RIN1*

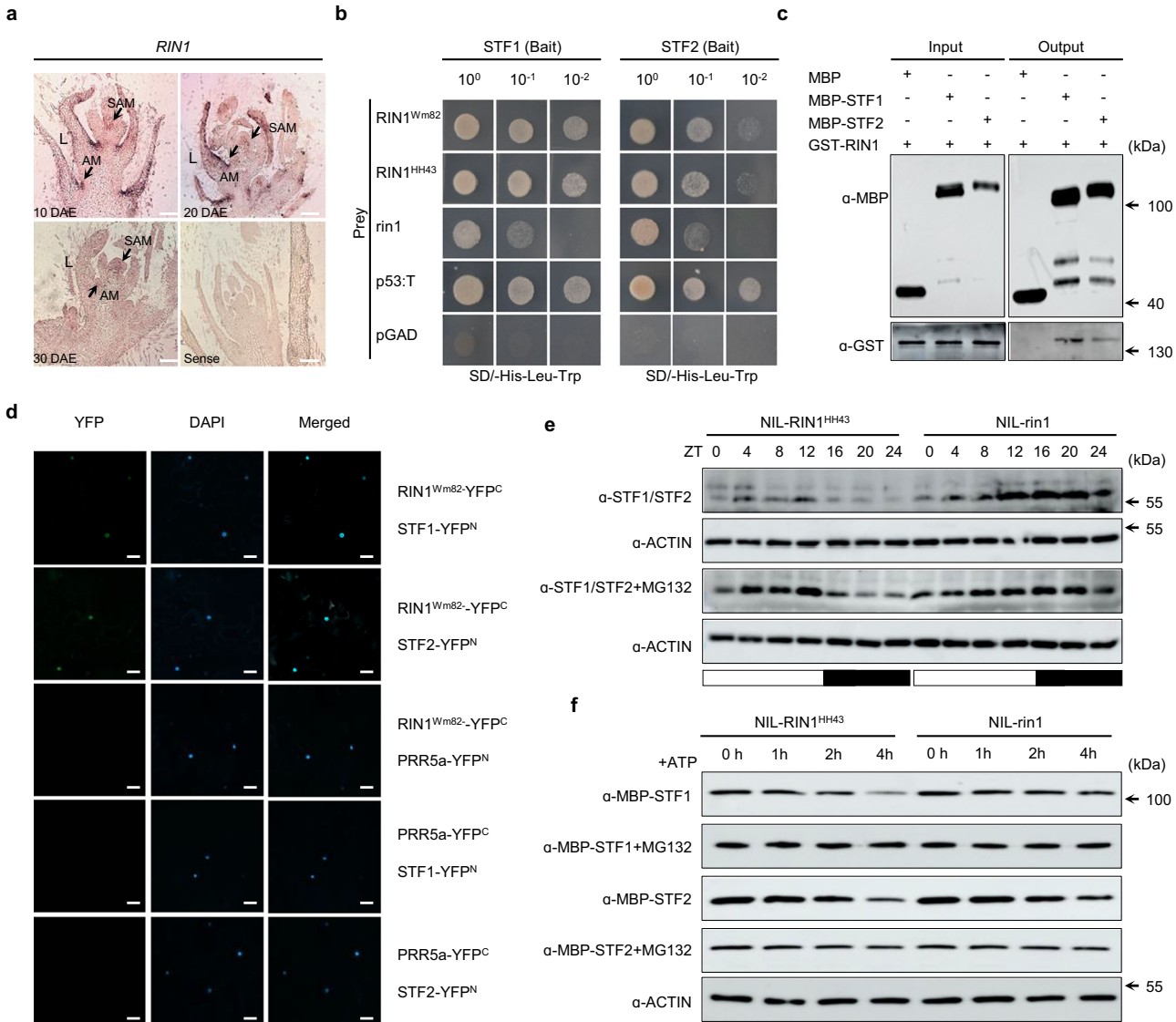

**Fig. 3 | RIN1 interacts with STF1 and STF2 and reduces STF abundance in vitro and in vivo. a** Spatial expression of *RIN1* in Wm82 stem tips at different periods as assayed by in situ hybridization. Arrows point to shoot apical meristems (SAM) and axillary meristems (AM). L: leaf primordium. Scale bars = 100 μm. Three independent biological replicates were performed. **b** Physical interactions between different genotypes of RIN1 and STF1 and STF2 in a yeast two-hybrid system. p53:T is a positive control and the empty pGAD vector is a negative control. **c** Pull-down assays verify the interaction between RIN1$^{Wm82}$ and STF1 and STF2 in vitro. Three independent biological replicates were performed. **d** BiFC analysis of physical interactions between RIN1$^{Wm82}$ and STF1 and STF2. YFP$^C$-tagged RIN1$^{Wm82}$ together with YFP$^N$-tagged STF1 and STF2, respectively, were transiently expressed in tobacco leaves. PRR5a-YFP$^C$ and PRR5a-YFP$^N$ were used for the negative controls. YFP$^C$: C-terminal YFP; YFP$^N$: N-terminal YFP. Scale bar = 20 μm. Three independent

biological replicates were performed. **e** RIN1$^{HH43}$ promotes the degradation of STF1 and STF2 in vivo, and STF1 and STF2 are stabilized in protein extracts from NIL-*rin1* plants compared to NIL-*RIN1*$^{HH43}$ plants in long-day photoperiods (16 h light/8 h dark). The membrane was probed with an antibody that recognizes both STF1 and STF2. Actin was used as a loading control. ZT: Zeitgeber time. Three independent biological replicates were performed. **f** In a cell-free degradation assay, RIN1$^{HH43}$ promotes the degradation of STF1 and STF2 in vitro, and both STF1–MBP and STF2–MBP are stabilized in protein extracts derived from NIL-*rin1* plants over NIL-*RIN1*$^{HH43}$ plants grown under long-day photoperiods (16 h light/8 h dark). Actin was used as a loading control. Three independent biological replicates were performed. Black arrows in (**c**–**e**, **f**) represent the positions of protein ladder. Source data are provided as a Source Data file.

expression in NIL-*RIN1*$^{HH43}$ and NIL-*rin1* over a 24-h period. From ZT0 to ZT24, *RIN1* expression was unchanged between two NILs (Supplementary Fig. 11e), indicating the *rin1* mutation does not reduce its transcriptional abundance. In summary, high *RIN1* expression in the shoot apical meristem at early phases of vegetative suggests the potential function in node fate and internode development, and thus contributes to plant height.

**RIN1 interacts with STF1 and STF2 and control STFs abundance**
In Arabidopsis, SPA proteins interact directly with HY5 through their WD40 domains and coiled-coil regions to control the accumulation of

HY5 and thus promote photomorphogenesis[17,22,27]. This prompted us to investigate whether RIN1/SPA3a might also interact directly with HY5 to promote the degradation of soybean HY5. Two soybean *HY5* homologs *STF1* (*Glyma.18G117100*) and *STF2* (*Glyma.08G302500*) mediate internode elongation by controlling the transcription of two *GA2ox7* homologs that determine the levels of bioactive gibberellins in soybean[16]. To this end, we performed yeast two-hybrid analysis (Y2H), a pull-down assay and bimolecular fluorescence complementation (BiFC) to investigate protein–protein interactions between RIN1 and STF1 and STF2. RIN1$^{Wm82}$ can interact with both STF1 and STF2 in vivo when expressed in yeast and tobacco, and in vitro (Fig. 3b–d).

To determine whether rin1 alters this interaction, we examined the interaction of RIN1[HH43] and rin1 with STF1 and STF2 in the GAL4 yeast two-hybrid system. rin1 can still interact with STF1 and STF2, but the interaction strength is weakened compared with full functional RIN1 (Fig. 3b). Because Arabidopsis SPAs can degrade their interaction partners via the 26 S proteasome, we next examined the abundance of STF1 and STF2 using an antibody that recognizes these STFs, in both NIL-RIN1[HH43] and NIL-rin1. STF1 and STF2 are less abundant in darkness but accumulate in the light. Meanwhile, RIN1[HH43] promotes the degradation of STF1 and STF2 both in the light and in darkness (Fig. 3e, Supplementary Fig. 12a). And in vitro, cell-free assay further supports the observation that protein extracts from NIL-RIN1[HH43] promote the degradation of STF1 and STF2, while the extracts from NIL-rin1 attenuate this degradation (Fig. 3f, Supplementary Fig. 12b, c). The addition of the proteasome inhibitor MG132 to the reaction decreased the protein degradation of STF1 and STF2 by RIN1 (Fig. 3e, f). To examine whether rin1 causes a change in STF expression, we quantified STF1 and STF2 transcription in NIL-RIN1[HH43] and NIL-rin1 over a 24-h period. rin1 mutations did not cause statistically significant changes on STF1 and STF2 transcriptions (Supplementary Fig. 13a, b). Taken together, RIN1 can interact directly with both STF1 and STF2 and promote degradation of STF1 and STF2, may also therefore control the expression of GA2ox7 genes to balance gibberellin levels and thus determine internode length in soybean as reported previously[16].

### RIN1 is genetically dependent on STF1 and STF2

To probe the genetic relationship between RIN1 and STFs, we generated a stf1 stf2 double mutant in the Wm82 background by CRISPR/Cas9 (Supplementary Fig. 14a, b). Compared with wild type, internode lengths for the stf1 stf2 double mutants are longer (Supplementary Fig. 14c, d). We further obtained stable transgenic STF2 over-expression lines, which have a dwarf phenotype (Supplementary Fig. 14e–h). These phenotypes are consistent with the previous results of STF1/STF2 from the genetic background under TL1 background[16] which further confirms that STF1 and STF2 participate in regulation of internode length in soybean. To confirm the genetic relationship between RIN1 and STF1 and STF2, we then backcrossed the NIL-rin1 with the stf1 stf2 double mutant in which the double mutant (in the Wm82 genetic background) was used as a recurrent parent for three generations to minimize background effects. We also developed a NIL set for RIN1 and STF1 and STF2 in the same genetic background. The phenotypic effect of rin1 significantly weakened in the stf1 stf2 mutant background (Supplementary Fig. 15), implying that RIN1 functions might be upstream of STF1 and STF2 and partially depends on STF1 and STF2.

### RIN1 regulates soybean plant height through gibberellin metabolism

A previous report showed GA2ox7a (Glyma.20G141200) and GA2ox7b (Glyma.11G003200) play roles in the CRY1–STF–gibberellin module to regulate low-blue-light-mediated plant height in soybean[16]. This implies that RIN1 may also affect internode length by regulating the expression of these GA2 oxidase genes. We therefore quantified the expression of GA2ox7a and GA2ox7b in NILs-RIN1 and wild-type Wm82, and rin1[CR1], rin1[CR2] at 10 DAE in shoot apical meristems wherein RIN1 is highly transcribed. GA2ox7a and GA2ox7b expression is up-regulated in NIL-rin1 and rin1[CR] over NIL-RIN1[HH43] and Wm82 controls, respectively (Fig. 4a, b). This indicates RIN1 represses the expression of GA2ox7a and GA2ox7b in shoot apical meristems. The same trends are reproduced by in situ hybridization in NILs-RIN1 for GA2ox7a and GA2ox7b in shoot-apical-meristem samples (Supplementary Fig. 16). To further explore the functional activity among different alleles of RIN1 to regulate the expression of GA2ox7a and GA2ox7b, we performed a dual-luciferase transient expression in tobacco leaves. The results

demonstrate that relative to the functional RIN1[Wm82] and RIN1[HH43], the inhibitory effect of rin1[CR1] on the promoter of GA2ox7a and GA2ox7b is completely lost, whereas the rin1 can induce the expression of GA2ox7a and GA2ox7b, although the ability is decreased (Fig. 4c, d). These results indicate that RIN1 can inhibit the expression levels of GA2ox7a and GA2ox7b, and rin1 might be a partial functional variant.

Introduction of the stf1 stf2 double mutant suppresses the expression levels of GA2ox7a and GA2ox7b, and over-expression of STF2 up-regulates GA2ox7a and GA2ox7b (Fig. 4e). This indicates RIN1 and STF1 and STF2 control GA2ox7a and GA2ox7b expression, to consequently determine internode length. Since RIN1 inhibits GA2ox7a and GA2ox7b expression while STF1 and STF2 promote its expression, we further examined the expression levels of GA2ox7a and GA2ox7b in RIN1 NILs on the background of STF1/STF2 and stf1/stf2, respectively. The result shows that although RIN1 can suppress the expression levels of GA2ox7a and GA2ox7b in both stf1/stf2 and STF1/STF2 backgrounds, the expression levels of GA2ox7a and GA2ox7b were significantly lower in the stf1/stf2 background than in the STF1/STF2 background (Supplementary Fig. 17), further confirming that RIN1 is partially dependent on STF1 and STF2. In addition, gibberellin levels in the shoot apices are elevated between the NILs of RIN1. As expected, the content of the bioactive gibberellin $GA_1$ is reduced and the content of the non-bioactive $GA_8$ is increased in the NIL-rin1 than the NIL-RIN1[HH43] (Fig. 4f), leading to shorter internodes and plant with rin1. Next, we treated shoot apical meristems of NIL-RIN1[HH43] and NIL-rin1 with different concentrations of $GA_3$ at V1 stage (after the first trifoliate leaf unrolled) and measured the internode length 1 week later. Exogenous $GA_3$ application reduced the internode-length differences between NIL-rin1 and NIL-RIN1[HH43], and restored the internode length of NIL-rin1 to that of NIL-RIN1[HH43] (Fig. 4g, h). Taken together, RIN1 regulates plant height and internode length by suppressing the expression of GA2ox7a and GA2ox7b, which in turn promotes gibberellin catabolism in the shoot apical meristem.

### rin1 improves soybean yield under high-density planting conditions

Our field-yield investigations suggest that NIL-rin1 reduces plant height, node number and internode length, but increases the total grains and grain weight per plant (Fig. 2e–h, Supplementary Fig. 18). The same trends were also observed with HN35 and its mutant rin1 (Supplementary Fig. 1). The rin1 allele confers a dwarf phenotype and improves yield, which could have potential to improve grain yield for soybean under dense planting conditions. To test this, we grew wild-type HN35 and its mutant rin1 at different planting densities in Harbin, China to evaluate yield, including 250,000, 350,000 and 450,000 plants per hectare. As the planting densities increased, plant height and internode lengths increased, and the grain yield per plant decreased (Fig. 5a–e). Regardless of planting densities used here, rin1 plants are shorter than wild-type HN35, but the grain yield per plant is higher than HN35 (Fig. 5a–e). Most strikingly, as planting density increases, the plot yield for rin1 is higher than HN35. At a density of 450,000 plants per hectare, the grain yield of rin1 is even higher still than that of HN35 (Fig. 5f). rin1 is therefore a desirable allele for improving soybean yield under dense planting conditions.

## Discussion

Owning to the rise of the semi-dwarf phenotype and green revolution, rice and wheat are grown densely with high nitrogen supplies, which significantly boosts their populational yield[30,31]. However, soybean dwarfism is often accompanied by fewer nodes and low yields. Identification of germplasm resources and genes suitable for high-density planting is essential for soybean yield improvement[32]. Here, we characterized the rin1 mutant that bears shorter internodes, moderately fewer nodes, and increased seed yield per plant (Supplementary Fig. 1).

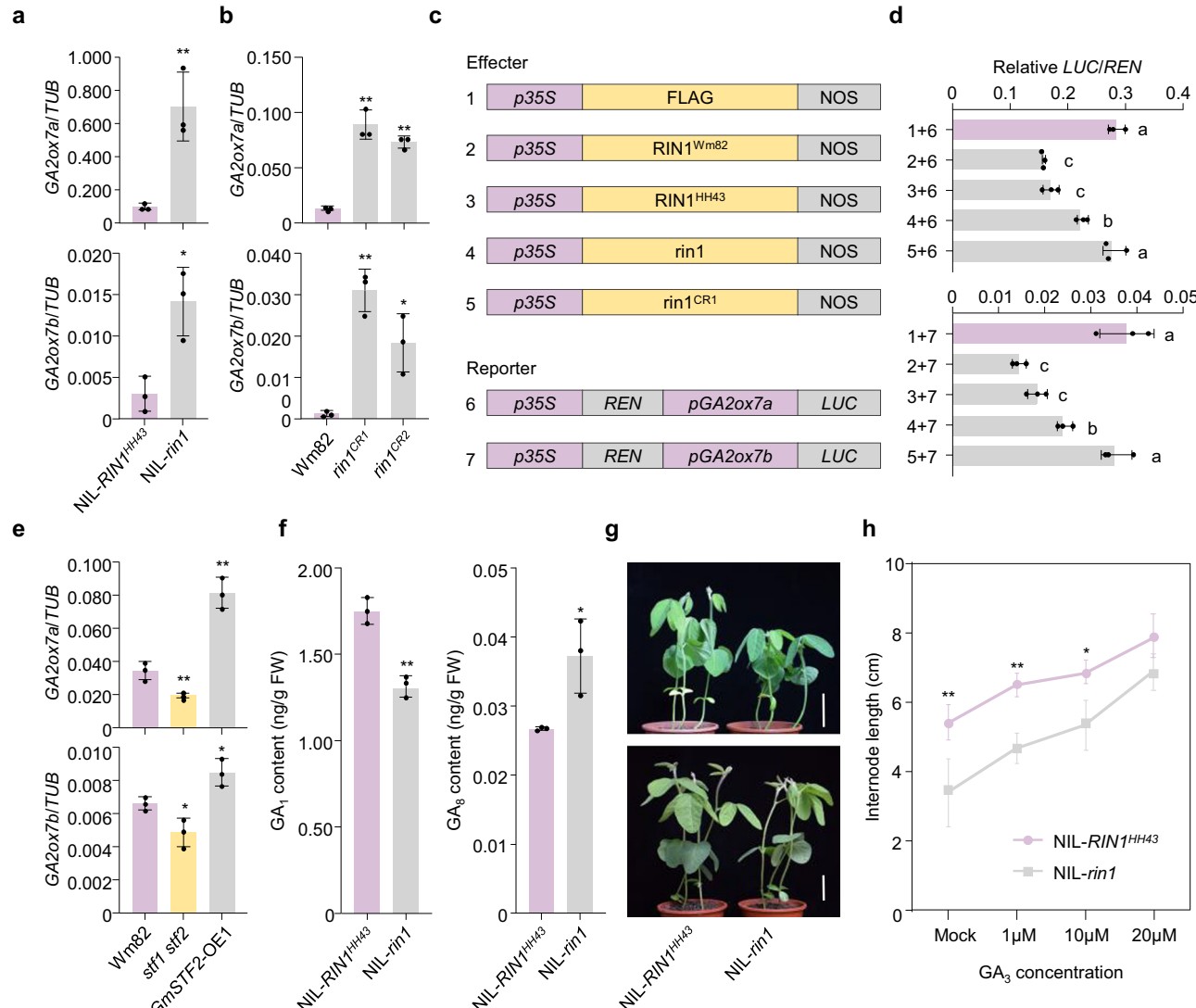

**Fig. 4 | *RIN1* regulates internode length through gibberellin metabolism.**
**a** Relative expression levels of *GA2ox7a* (top) and *GA2ox7b* (bottom) in NIL-*RIN1*[HH43]
and NIL-*rin1* under LD conditions (16 h light/8 h dark) in a growth chamber.
**b** Relative expression levels of the *GA2ox7a* (top) and *GA2ox7b* (bottom) genes in
Wm82 and *rin1*[CR1], *rin1*[CR2] mutants under LD conditions (16 h light/8 h dark) in a
growth chamber. Data shown in (**a**, **b**) are relative to the control gene *Tubulin* (*TUB*).
Data shown in (**a**, **b**) are means ± SD from three independent biological replicates. A
student's *t*-test (two-sided) was used to generate the *P*-values of (**a**, **b**), *$P < 0.05$,
**$P < 0.01$. **c** Constructs used for the transient transfection assay. *p35S*, *35S* Pro-
moter. *pGA2ox7a*, *GA2ox7a* Promoter. *pGA2ox7b*, *GA2ox7b* Promoter. **d** *Luciferase*
(*LUC*) activity under the control of the *GA2ox7a* or *GA2ox7b* promoter, the com-
binations in (**d**) link to the numbers in (**c**). Data are means ± SD from three inde-
pendent biological replicates. One-way ANOVA with Tukey's post-test was
performed to determine statistically significant differences, $P < 0.05$. **a**–**c** letters
indicate statistically significant differences. **e** Relative expression levels of the
*GA2ox7a* (top) and *GA2ox7b* (bottom) genes in Wm82, *stf1 stf2* double mutants and

*GmSTF2-OE1* under LD conditions (16 h light/8 h dark) in growth chamber. Data
shown are relative to the control gene *Tubulin* (*TUB*). Data shown are relative to the
control gene *Tubulin* (*TUB*). Data shown are means ± SD from three independent
biological replicates. A student's *t*-test (two-sided) was used to generate the *P*-
values, *$P < 0.05$, **$P < 0.01$. All the NIL-*RIN1*[HH43] and NIL-*rin1*, Wm82, *stf1 stf2* double
mutants and *GmSTF2-OE1* lines used for RT-qPCR in (**a**, **b**–**e**) were taken from the
stem tips of seedlings at 10 DAE under LD incubators (16 h light /8 h dark). **f** GA$_1$
(left) and GA$_8$ (right) contents in NIL-*RIN1*[HH43] and NIL-*rin1* backgrounds determined
by LC–MS/MS. Data shown are means ± SD from three independent biological
replicates. A student's *t*-test (two-sided) was used to generate the *P*-values,
*$P < 0.05$, **$P < 0.01$. **g** Phenotypes of NIL-*RIN1*[HH43] and NIL-*rin1* seedlings treated
with 20 μM GA$_3$ at the V1 stage (one unrolled trifoliate leaf) (top) and after a week
(bottom). Scale bar = 5 cm. **h** Internode lengths of seedlings treated with different
concentrations of GA$_3$. Data shown are means ± SD from three independent bio-
logical replicates. A student's *t*-test (two-sided) was used to generate the *P*-values,
*$P < 0.05$, **$P < 0.01$. Source data are provided as a Source Data file.

*RIN1* is encoded by *SPA3a* (Fig. 1) and the RIN1 protein interacts with
STF1 and STF2 and regulates STF abundance both in vitro and in vivo
(Fig. 3). Furthermore, expression of *GA2ox7a* and *GA2ox7b* is regulated
by RIN1–STF module to control internode elongation through balan-
cing bioactive and inactive gibberellin pools (Fig. 4). Our study not
only identifies a short-internode mutant for increasing soybean yield
(Fig. 5), but also provides a mechanistic understanding of the
RIN1–STF–gibberellin module for regulation of soybean internode
length (Fig. 6).

The Green Revolution in the 1960's has led to the doubling of the
grain yield of rice and wheat[10,30,31]. This is mainly due to beneficial semi-
dwarf plant architecture, which improves the response to high fertili-
zer inputs and is accompanied by lodging resistance, enhanced light
use, and tolerance of high-density planting − the combination of which
has led to substantial increases in grain production[32,33]. In contrast to
the significant increases in yield for rice, wheat and maize, soybean
yield has not improved significantly over the past few decades, sug-
gesting soybean improvement was left behind by the Green

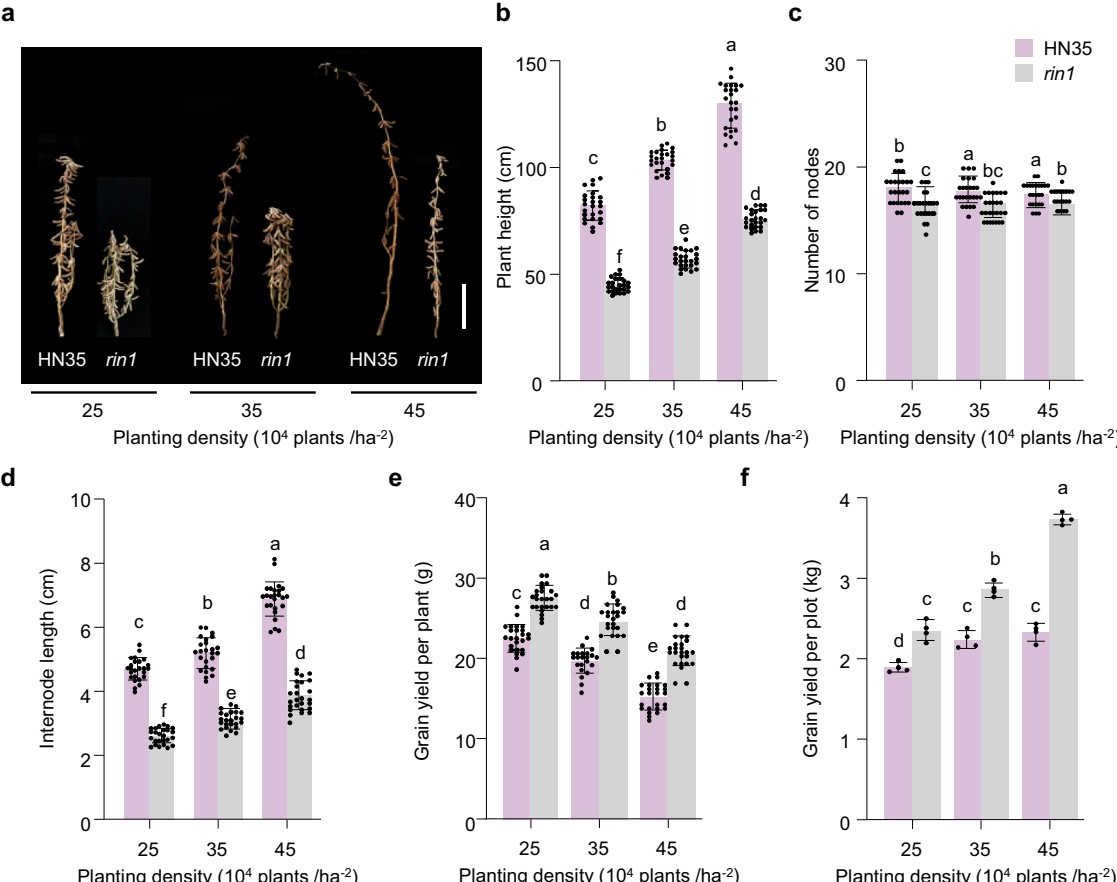

**Fig. 5 | *rin1* enhances yield under different planting densities under field conditions. a–f** Phenotype chacterization of wild-type HN35 and *rin1* mutants planted at different densities (25, 35 or 45 × 10⁴ plants/ha⁻²) in a nature field in Harbin, China (45°75′N, 126°63′E). **a** Phenotype of wild-type HN35 and *rin1* mutants planted at different densities in the field. Scale bar = 20 cm. **b** Plant height (cm), *n* = 25.

**c** Number of nodes, *n* = 25. **d** Internode length (cm), *n* = 25. **e** Grain yield per plant (g) *n* = 25. **f** Grain yield per plot (kg), *n* = 4. Data are means ± SEM. One-way ANOVA with Tukey's post-test was performed to determine the effect of planting density on the indicated yield traits, *P* < 0.05. **a–f** letters indicate statistically significant differences. Source data are provided as a Source Data file.

Revolution[34]. Although the Green Revolution greatly increased crop yields, this requires high nitrogen inputs to achieve such yield[30,35]. However, soybean possesses the ability of a symbiosis to fix atmospheric nitrogen in root nodules containing rhizobia when soil nitrogen is limited, and >70% of nitrogen required for soybean growth is supplied from symbiosis. High concentrations of inorganic nitrogen in the soil inhibits nodule formation and nitrogen fixation, thus inhibiting symbiotic nitrogen fixation and tremendously reducing grain yield[36–39]. Therefore, unlike cereal crops, the harmonization of nitrogen fixation in nodules with nitrogen supply is essential for enhancing soybean yield, and therefore overcoming this balance may mean it is more difficult than with cereal crops to achieve yield increases. Nonetheless, the *rin1* allele isolated here clearly plays an important role in conferring dwarfism to soybeans and facilitates high-density planting, but whether it contributes to nodulation and nitrogen fixation under low-nitrogen conditions needs further evaluation.

As a photoperiod-sensitive, short-day crop, maturity of soybean also has a major impact on soybean yield at different latitudes[28,40]. At high latitudes, photoperiods are longer, which requires early-maturing soybean cultivars with reduced photoperiod sensitivity. However, early maturity frequently leads to a shortened growth period and is accompanied by low grain yield[41]. Achieving early maturity without a yield penalty is an remains an important breeding objective for production at high latitudes—the major soybean production regions in China and North America[41–43].

*SPA* genes are known to regulate flowering time in Arabidopsis[44–49]. In contrast to wild-type soybean SPA3 proteins, our

CRISPR knockout mutants *rin1^CR1* and *rin1^CR2* lack the kinase domain, coiled-coil domain and WD40-repeat domain, and flower earlier than Wm82 under non-inductive, long-day photoperiods (Supplementary Fig. 10a, b), alongside roles internode development, indicating that *RIN1* is also involved in soybean flowering. Variation in flowering time in natural populations is also observed in which *RIN1*-H1 and *RIN1*-H3 flower earlier than *RIN1*-H2 (Supplementary Fig. 9h–j). However, there are statistically non-significant differences in flowering time between NIL-*RIN1^HH43* and NIL-*rin1* (Supplementary Fig. 10c, d). There may be two explanations for this. One is that the genetic backgrounds are different and some key factors/genes required for *RIN1* functions may be not functional. The other explanation may be that the *rin1* mutant has only a partial truncation of the WD-domain, but has a functional kinase domain and coiled-coil domain, of which these two domains are potentially sufficient to function in flowering-time regulation, yet are unable to control internode development. And through transient experiments, we also conject that *rin1^CR* is a complete loss-of-function mutant while *rin1* is a partial loss-of-function mutant. These observations suggest that the molecular mechanisms underlying *RIN1*-mediated flowering time and internode length are independent.

In this report, we identified or generated six alleles of *RIN1*, namely *RIN1*-H1, *RIN1*-H2, *RIN1*-H3, *rin1^CR1*, *rin1^CR2* and *rin1*. *RIN1*-H1, *RIN1*-H2 and *RIN1*-H3 haplotype groups do not differ in internode length and plant height, but do differ in flowering time, suggesting that they may be useful for creating bespoke, locally adapted soybean cultivars. Interestingly, the knockout alleles of *rin1^CR1* and *rin1^CR2* not only reduce plant height and internode length, but also promote flowering. Whether

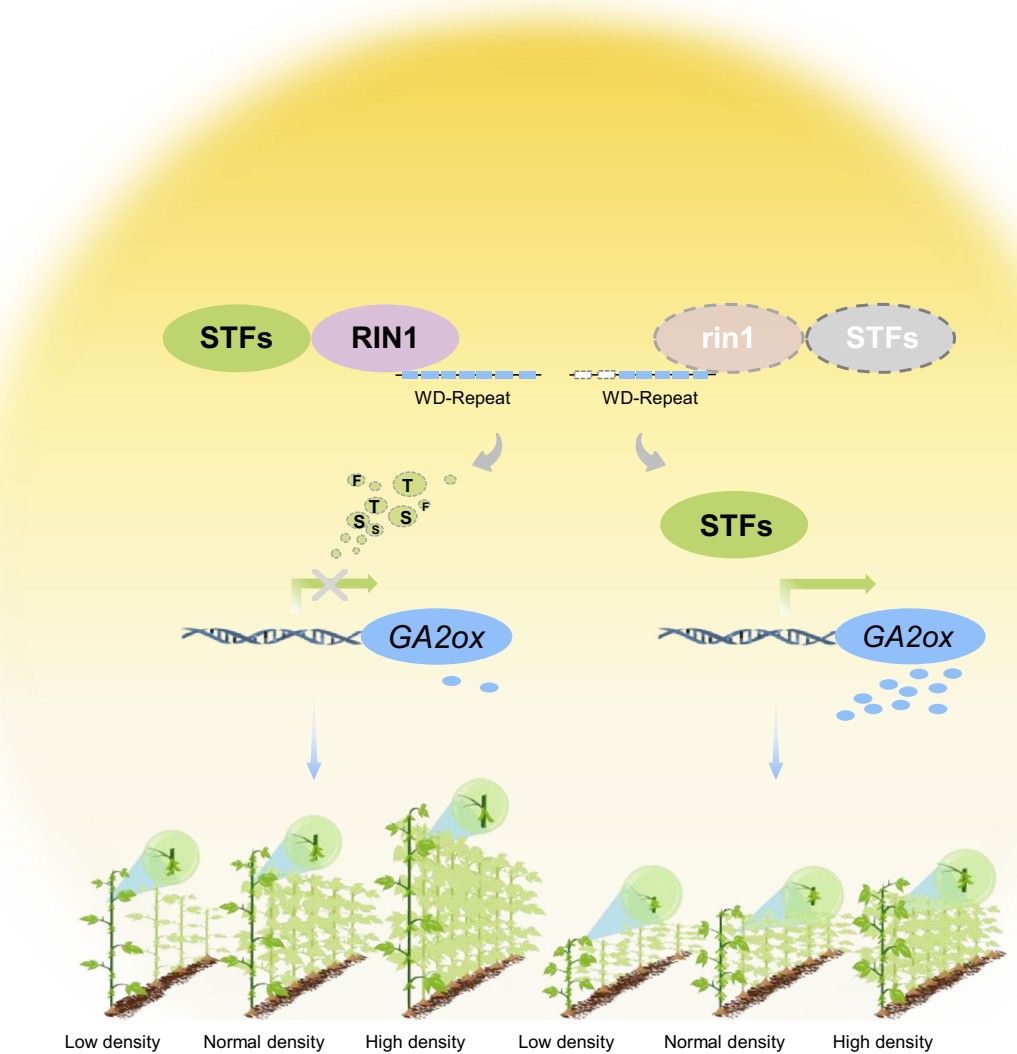

**Fig. 6 | Proposed model of *RIN1* function in regulating internode length and the potential utility of *rin1* for high-density planting under long-day photoperiods.** RIN1 interacts with STF1 and STF2 and decrease STF protein abundance. This leads to down-regulation of *GA2ox* expression to promote internode elongation. While the truncation of WD40-repeat domain in rin1 weakens but does not eliminate the interaction with STF1 and STF2, and leads to an increase in STF abundance to directly up-regulate expression of *GA2ox*[16]. This results in a shortened internode. With increasing planting densities, yield of *rin1* increases, thus demonstrating *rin1* is an elite allele for dense-planting regimes typical of contemporary soybean-production systems.

both knockout alleles can improve soybean grain yield under dense planting conditions with early maturity remains to be clarified, though they may be key alleles to overcome the tradeoff between early maturity and low grain yield. While *rin1* mutants do not affect flowering time, but have greatly increased yield under high-density planting environments (Fig. 6), introgression of *rin1* into elite cultivars is a promising strategy to rapidly improve soybean yield under dense planting conditions, with potential use in soybean–maize intercropping systems without changing their maturity.

## Methods

### Plant materials and phenotyping

Selection of the population parents: Heinong 35 (HN35) and Heihe43 (HH43) are elite cultivars widely grown in northeast China. The soybean line *rin1* was obtained by screening a γ-ray mutant library of HN35. HN35, HH43 and *rin1* were planted under natural long-day (NLD) conditions in a field in Harbin, China (45°75′N, 126°63′E). These plants were sown at the beginning of May and harvested in October, 2017. Each row was 2 m long, with a spacing of 60 cm between rows. About

20 plants were sown in each row, and three rows were planted for each variety.

QTL detection: HH43 was crossed with *rin1* to obtain the $F_2$ population (HNH population), consisting of 207 progenies. In 2017, all 207 progenies were planted under NLD conditions in the same field in Harbin. Each row was 2 m long, with 60 cm spacing between rows. 20 plants were sown in each row at the beginning of May and harvested in October. After harvest, plant height and node number of each plant was recorded, and the average internode lengths were calculated.

In 2018, HN35, HH43, *rin1* and $F_{2:3}$ population were grown in the same field in Harbin. For the $F_{2:3}$ population, we randomly selected 20 seeds from each harvested $F_2$ plant and planted each $F_2$ progeny in a row. Each row was 2 m long with 60 cm spacing between rows. These plants were sown at the beginning of May and harvested in October. After harvest, internode length in each row of $F_{2:3}$ was calculated and used as a replicate of $F_2$.

Fine mapping: Soybean plants from the $F_4$ to $F_6$ population and residual heterozygous lines (RHLs) were grown in Harbin from 2019 to 2021. Plants were sown at the beginning of May and harvested in

October of each year. Each row was 2 m long with 60 cm spacing of between rows. About 20 plants were sown in each row.

Differential planting-density experiments: Four replicates were performed in the nature field of Harbin in 2020. Plants were sown in the middle of May, with three different plant densities (250,000, 350,000, and 450,000 plants per hectare). For 250,000 and 350,000 plants per hectare, seeds were only set in a row, and the average spacing between plants was 6.08 cm and 4.34 cm, respectively. For 450,000 plants per hectare, the row spacing remained at 60 cm. While the seeds were set in two rows with a zigzag shape, and the average spacing between plants was narrowed down to 3.38 cm. Plant height, number of nodes, internode length, grain yield per plant and grain yield per plot were recorded and calculated after harvest.

## Map-based cloning and NIL construction

QTL detection and the logarithm of odds (LOD) threshold were achieved with MapQTL5 software. Fine mapping of *RIN1* using in a segregating-heterozygous inbred family that was heterozygous only at *RIN1*, eight Indel markers were identified (Fig. 1d), and internode length phenotype was used to evaluate the genomic interval containing *RIN1*. Primer sequences of the markers used for fine-mapping are listed in Supplementary Table 4.

NILs for the *RIN1* locus were selected from the $F_6$ progeny. The NIL possessed the *rin1* allele from *rin1* mutant and the *RIN1* allele was sourced from HH43 (Supplementary Fig. 8). NILs were grown in Shi-jiazhuang, China (37°27′N, 113°30′E) with a NLD conditions in June 2022 and harvested in October. Each row was 2 m long, with 60 cm spacing between rows. About 30 plants were sown per row. The phenotypes mentioned above were all recorded: plant height, number of nodes, number of pods per plant, number of grains per plant, and yield per plant after maturity. The insertion or deletion (Indel) and single-nucleotide polymorphism (SNP) markers (Supplementary Table 4) were developed to map-based cloning and NILs construction.

## DNA extraction

Part of a fresh trifoliate leaf was harvested then stored at −80 °C. Genomic DNA was extracted using the NuClean PlantGen DNA Kit (CWBIO, Beijing, China). DNA quality was ascertained by electrophoresis of an aliquot on a 1% agarose gel, and DNA concentration was determined by NanoDrop 2000 (ThermoScientific, Wilmington, DE, United States).

## RNA extraction and RT-qPCR

All plants used were grown in a growth chamber under long-day photoperiods (16 h light/8 h dark), 25 °C, and the trifoliate leaves were sampled at 10 DAE at Zeitgeber time (ZT) 0, 4, 8, 12, 16, 20 and 24. Total RNA was extracted from the shoot apices and trifoliate leaves of NILs, Wm82, *rin1*$^{CR1}$ and *rin1*$^{CR2}$ mutants with the Ultrapure RNA Kit (CWBIO, China). RNA extractions were repeated three times using the shoot apices planted in different long-day conditions (16 h light/8 h dark). First-strand cDNAs were synthesized from RNA using the PrimeScript RT Reagent Kit with genomic DNA Eraser (Takara, Japan) and their concentrations determined with a NanoDrop 2000 instrument (ThermoScientific, Wilmington, DE, United States). The TB Green® Premix Ex Taq™ II (Takara) was used as the master mix. The LightCycler 480 instrument (Roche) was used for RT-qPCR. *Tubulin* (*TUB, Glyma.05G157300*) was used as reference for normalization and the corresponding primers are listed in Supplementary Table 6. Three biological replicates were used in all assays, and each sample was analyzed in technical triplicate.

## Haplotype calling

The resequencing data of the 1,295 accessions queried in this study were obtained from Lu et al., 2020[28]. The panel consists of 146 wild accessions, 575 landraces and 574 improved cultivars, which are listed in Supplementary Table 5. Individuals with poor sequencing data for *RIN1* were removed from the haplotype analysis, and finally 1,095 accessions were used for haplotype calling. The 297 accessions consisted of 137 landraces and 160 improved cultivars of these accessions were used for association analysis. VCFtools was used to process VCF files. The statistical significance of association was calculated with by IBM SPSS software version 26 (https://www.ibm.com/spss). One-way ANOVA with Tukey's post-test was performed to determine statistically significant differences, $P < 0.05$. *Glycine max* Wm82.a2.v1 was used as the reference genome for annotation[50,51].

## Construction of phylogenetic trees

The amino-acid sequences of GmSPAs, AtSPAs, OsSPA3 and ZmSPA3 were obtained from Phytozome (https://phytozome.jgi.doe.gov/pz/portal.html), and amino-acid sequences of PsSPA3, VuSPA3, VuSPA3a and PvSPA3 were from Legume Information System (LIS) (https://www.legumeinfo.org). MEGA6 software was used to construct the phylogenetic trees by neighbor joining method.

## Vector construction and soybean transformation

The vector used for CRISPR/Cas9 was from Ma et al., 2015[52]. The targets were designed by CRISPRdirect (http://crispr.dbcls.jp/) and are shown in Supplementary Fig. 4a, primers used for plasmid construction are listed in Supplementary Table 6. The knockout construct was introduced into the CRISPR/Cas9 plasmid and then transformed into Wm82 plants[53]. The *RIN1* CRISPR/Cas9-edited plants were identified by selection on phosphinothricin and confirmed by PCR and sequencing.

The full-length CDS of *STF2* was amplified from cDNA obtained from Wm82 leaves ligated with *p35S–pTF101–3FLAG* vector and introduced into *Agrobacterium* strain EHA101 to give rise to the *p35S–STF2–3FLAG* vector. This was transformed into soybean cultivar Wm82. Western blotting against the FLAG epitope was used to detect transgene expression in the transgenic lines. Primers used for plasmid construction are listed in Supplementary Table 6.

## Yeast two-hybrid assays

The yeast two-hybrid assay was conducted with GAL4 system. The full-length CDS of *RIN1*, *STF1* and *STF2* were amplified from cDNA obtained from leaves of Wm82. The full-length CDS of *RIN1*$^{HH43}$ and *rin1* were amplified from HH43 and the *rin1* mutant, respectively. The plasmids containing RIN1$^{Wm82}$, RIN1$^{HH43}$ and rin1 were introduced into the prey vector pGADT7 and the plasmids with STF1 and STF2 were introduced to the bait vector pGBKT7. Next, equal amounts of paired plasmids were cotransformed into the yeast strain Y2H Gold (Clontech) and grown on SD/-Leu-Trp solid medium. Cultures were incubated at 30 °C for 3 days with inversion. Approximately equal amounts of clones were selected and dotted on SD/-His-Leu-Trp solid medium.

## Bimolecular fluorescence complementation (BiFC)

To prepare vectors for the BiFC assay, a plasmid containing *RIN1*$^{Wm82}$ and *PRR5a* (*Glyma.06G136600*) were cloned into the pUC–SPYCE vector and the plasmids containing *STF1*, *STF2* and *PRR5a* were cloned into the pUC–SPYNE vector to generate pYFP$^C$–RIN1, pYFP$^C$–PRR5a, pYFP$^N$–STF1, pYFP$^N$–STF2 and pYFP$^N$–PRR5a. The constructs were transformed into *A. tumefaciens* strain GV3101. A single clone was selected and was used to inoculate liquid LB containing spectinomycin and rifampicin selection, then incubated at 28 °C until OD$_{600\ nm}$ = 0.6–0.8. After centrifugation at 2,000 x *g* for 5 min, bacteria were pelleted then resuspended in resuspension solution (10 mM MgCl$_2$, 10 mM MES, 150 μM acetosyringone, pH = 5.6). Cells were maintained at 25 °C in darkness for 1.5 h. Two strains of Agrobacteria of interest were mixed in equal amounts and infiltrated into tobacco leaves (*Nicotiana benthamiana*). Infiltrated tobacco plants were maintained in long-day conditions (14 h light/10 h dark) for about

2 days. YFP fluorescence was observed with a confocal laser-scanning microscope (Zeiss).

## In vitro pull-down assay

RIN1[Wm82] and STF1 and STF2 coding sequences were inserted into pMAL–c5x and pGEX4T–1 vectors, respectively, and transformed into *E. coli* BL21, then purify the proteins with pull down buffer. 10 µL MBP beads were incubated with 10 µg MBP–STF1/STF2 for 1 h at 4 °C, and then 1 µg GST–RIN1 was added for 1 h. Beads were centrifuged, the supernatant removed, and beads were washed 5 times. Finally, SDS–PAGE loading buffer was added to samples for denaturation by boiling. Western blotting was used to verify the interaction.

## In situ hybridization

Shoot apices were dissected from Wm82 at 10, 20 and 30 DAE from plants grown under long-day photoperiod (16 h light/8 h dark), 25 °C. Shoot apices were harvested from NILs, WT (Wm82) and *rin1*[CR1] at 10 DAE from plants grown under long-day photoperiod (16 h light/8 h dark), 25 °C. Samples were immediately put into 3.7% v/v FAA (70% v/v ethyl alcohol, 5% v/v acetic acid, and 2% v/v formalin)[54] for storage. The preserved samples were embedded in paraffin wax, sectioned with a microtome, and hybridized cross with the probes. The primers to make specific probes of *RIN1*, *SPA3b*, *GA2ox7a* and *GA2ox7b* are listed in Supplementary Table 6.

## Cell-free in vitro degradation assays

Shoot apices sampled from NIL-*RIN1*[HH43] and NIL-*rin1* were used for cell-free protein-degradation assay. NIL-*RIN1*[HH43] and NIL-*rin1* plants were grown under long-day photoperiod (16 h light/8 h dark) and the shoot apices were harvested from 10-day-old plants at ZT0, 4, 8, 12, 16, 20, and 24. Each sample of ZT0–ZT24 were mixed and protein samples were extracted with extract solution buffer (25 mM Tris–HCl pH 7.5, 10 mM $MgCl_2$, 10 mM NaCl, 5 mM DTT, 1 mM PMSF, 10 mM ATP, and 100 µM cycloheximide)[55]. In the MG132-added samples, the final concentration of MG132 was 40 µM. Equal amounts of STF1–MBP and STF2–MBP were added to each sample and incubated at room temperature for 0 h, 1 h, 2 h, and 4 h. SDS–PAGE loading buffer was then added for denaturation by boiling immediately. Western blotting was performed to determine protein abundance. Actin was used as a loading control.

## In vivo degradation assays

NIL-*RIN1*[HH43] and NIL-*rin1* plants were grown under long-day photoperiod (16 h light/8 h dark) and shoot-apex samples were harvested from 10-day-old plants at ZT0, 4, 8, 12, 16, 20 and 24. For the samples containing MG132, seedlings were treated with 10 µM MG132 2 days before sampling. Proteins were extracted separately for each sample from ZT0 to ZT24. SDS–PAGE loading buffer was added for denaturation by boiling. Western blotting with the STF1/STF2 antibody was performed to investigate degradation. The STF1/STF2 antibody was obtained from Lyu et al., 2021[16]. Actin was used as a loading control.

## Transient expression assay

To generate the *pGA2ox7a–LUC* and *pGA2ox7b–LUC* constructs, about 2 kb promoter sequences of *GA2ox7a* and *GA2ox7b* were amplified from Wm82 and introduced into the pGreenII *O800–LUC* vector. The different alleles of *RIN1* (*RIN1*, *RIN1*[HH43], *rin1* and *rin1*[CR1]) were introduced into the p35S–pTF101–3FLAG vector to generate the constructs *p35S-RIN1–3FLAG*, *p35S-RIN1*[HH43]*–3FLAG*, *p35S-rin1–3FLAG*, *p35S-rin1*[CR]*–3FLAG*. The *pGA2ox7a–LUC* and *pGA2ox7b–LUC* constructs were used as the reporters while the various RIN1 constructs were used as the effectors in the tobacco transient expression system. Two strains of Agrobacteria of interest were mixed in equal amounts and infiltrated into tobacco leaves (*Nicotiana benthamiana*). Infiltrated tobacco plants were maintained in long-day conditions (14 h light/10 h dark) for about 2 days. The *LUC* and *REN* activities were measured

under the manufacturers' instructions (Promega). The *LUC/REN* ratio was presented with three biological replicates, and the primers used are listed in Supplementary Table 6.

## Gibberellin quantification

To quantify gibberellin profiles, NIL-*RIN1*[HH43] and NIL-*rin1* seedlings were grown under long-day photoperiod (16 h light/8 h dark). Shoot apices were harvested at 10 DAE and stored at −80 °C. Samples were homogenized into a powder and then added 10 µL internal standard mixed working solution with a concentration of 10 ng/mL. The extraction was carried out by vertexing 1 mL methanol: water: formic acid (15:4:1, v/v/v) solution for 15 min, and the supernatant was taken after centrifugation at 5000 x *g* for 10 min at 4 °C. Concentrated the supernatant to dryness, added 3.5% formic acid and ethyl acetate, vortexed for 10 min, centrifuged at 5000 x *g* for 5 min at 4 °C, and taken the upper layer of ethyl acetate. Dried the extract with nitrogen, redissolved in acetonitrile/water (90:10, v/v) solution, and passed through a 0.22 µm filter membrane. Liquid chromatography-tandem mass spectrometry (LC-MS/MS) was used to analyze. The procedure of LC-MS/MS was based on previous studies with slight modification[56,57]. The data acquisition instrument system mainly includes Ultra Performance Liquid Chromatography (UPLC) (ExionLC™ AD) and Tandem Mass Spectrometry (MS/MS) (QTRAP® 6500 + ). Mobile phase: Phase A, 0.05% formic acid/ultrapure water; Phase B, 0.05% formic acid/acetonitrile. The Gradient elution procedures are as follows: 0 min A/B was 95:5 (V/V), 10 min A/B was 5:95 (V/V), 11 min was 5:95 (V/V), 11.1 min A/Bwas95:5 (V/V) and 14 min was 95:5 (V/V). Flow rate 0.35 mL/min; Column temperature 40 °C and the injection volume was 10 µL. Three biological replicates were performed.

## Reporting summary

Further information on research design is available in the Nature Portfolio Reporting Summary linked to this article.

## Data availability

Data supporting the findings of this work are available within the paper and its Supplementary Information files. The sequencing data used in this study were previously reported were deposited into the NCBI database under accession number SRA: SRP045129 and deposited into the Genome Sequence Archive (GSA) database in BIG Data Center (http://gsa.big.ac.cn/index.jsp) under Accession Number PRJCA001685. The Wm82 a2.v1 reference genome was download from *Phytozome*. Source data are provided with this paper.

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

## Acknowledgements

This work was supported by the National Key Research and Development Program (Nos. 2021YFF1001100 to S.Lu), National Natural Science Foundation of China (32090064 to F.K., 32201865 to S.Li., 32022062 to S.Lu.), the National Key Research and Development Program (Nos. 2022YFD1201501 to F.K.), and the Major Program of Guangdong Basic and Applied Research 2019B030302006 to F.K. and Ba.L., the open competition program of top ten critical priorities of Agricultural Science and Technology Innovation for the 14th 5-Year Plan of Guangdong Province (2022SDZG05) to F.K. and Ba.L., and National Key Research and Development Program (Nos. 2021YFF1001203 to Ba. L.).

## Author contributions

F.K., Ba.L., and S.Lu. designed and supervised the experiments and managed the projects. F.K. and Bi.L. interpreted the results. Ba.L. generated and screened for rin1 mutants. X.Fu. generated the stf1 stf2 double mutant. C.Q. and B.L. generated the STF1/STF2 antibody. S. Li, Z.S., and X.H. performed the biochemical experiments. S.Li. and L.K. fine-mapped the RIN1 locus. Hu.L. performed the RNA in situ hybridization. T.S., H.Li., M.He., L.W., and S.Liu. constructed part vectors and performed the transformation experiments. S.Li., C.F. and S.Lu. performed the data analysis. S.Li. and S.Lu. drafted the manuscript. F.K., Q.S., and B.L. revised the manuscript.

## Competing interests

The authors declare no competing interests
