## [Peer Review File · Nature Communications]

Soybean REDUCED INTERNODE 1 determines internode length
and improves grain yield at dense plantingReviewer #1 (Remarks to the Author):

The authors report results that can be very useful for increasing overall seed yield of soybean. The manuscript is presented well and logically. However, the English language in the manuscript needs improvement. I have made some edits (highlighted in the attached pdf), but more edits by an Editor will help improving the overall quality of the manuscript. I have few comments inserted in the manuscript. Specifically.

Lines 210-11, the sentence is not clear to me

Lines 346-348, how many replicates? 20 plants in 2 m is a thin planting, normal is ~30 plants.

Lines 354-55: mention what happened to row spacing and plant spacing within row with increase in planting density.

I recommend publication of this manuscript after suggested revision.

In their review of the first version of this manuscript, reviewer #1 added some comments to the manuscript file. These comments were forwarded to the authors, who replied as included in this Peer Review File.

Reviewer #2 (Remarks to the Author):

Li et al. describe the identification of a gene, RIN1, involved in the determination of plant height and internode length in soybean. After random mutagenesis and screening for plants with reduced height, they isolate a mutant (*rin1*) bearing a premature termination codon in a gene that encodes a SUPPRESSOR OF PHYA (SPA) protein, that belongs to a family of ubiquitin ligases involved in plant photomorphogenesis and other aspects of development. Interestingly, plants with this particular allele show increased seed yield (relative to that of plants with a functional RIN1 gene) when planted at high density, which allows higher seed production per unit area. The authors advance in studying the molecular mechanism of RIN1 action. They show that, like its counterpart from Arabidopsis, it interacts with the soybean HY5 homologues STF1 and 2 and reduces the abundance of these transcription factors, possibly by mediating their degradation. Since previous data from the literature indicate that STFs control internode elongation acting on the expression of two gibberellin (GA) catabolism genes, they measure expression of these genes and GA levels and show that they are increased and decreased, respectively, in plants bearing the *rin1* allele. They conclude that RIN1 controls plant height and internode length through STF1/2 and GA levels and that the *rin1* allele they identified could be a useful tool to improve grain yield in soybean under high density planting conditions.

The results are interesting considering their potential application in agriculture for the development of soybean varieties with increased yield. Regarding the molecular mechanism of RIN1 action, the novel aspect is that RIN1 affects STF1/2 stability and apparently acts through STF1/2 to modulate internode length, while the involvement of STF1/2 in the determination of plant height and their role in the regulation of GA catabolism genes were already known. In addition, it is also known that the STF1/2 homologue from Arabidopsis, HY5, is a substrate of SPA proteins that target it for degradation, so the mechanism seems to be analogous to the one observed in Arabidopsis. That said, it is relevant that mutation of this particular SPA protein from soybean (RIN1) affects internode length, since the soybean genome encodes 10 different SPA proteins.

Particularly, I find that the exact molecular mechanism involved in the action of RIN1 is not completely clear and additional experimentation is required to fully support the conclusions raised by the authors. In addition, additional controls are required in specific experiments to support the results.

Finally, while the manuscript is understandable, a thorough revision of language is required to meet the standards of the journal.

Specific comments:

-To show that the mutation of RIN1 is the cause of the shorter internode length, the authors obtained two CRISPR/Cas9 derived mutants with stop codons near the N-terminus of the protein (unlike the isolated *rin1* allele, that contains a stop codon near the C-terminus). They also obtain near-isogenic lines that bear the HH43 (wild-type) and the *rin1* mutant allele. Through this, they confirm that RIN1 loss-of-function affects plant size and internode length. Since the authors link these parameters with grain yield, it would be interesting to know if these plants also show an increase in this parameter. In fact, in the last sentence of Results the authors conclude that "These results indicate that *rin1* is an ideal allele for improving soybean yield under dense planting conditions." Is this specific for the *rin1* allele or knocking out RIN1 as in the CRISPR mutants has a similar effect? This also leads to the question of the nature of the *rin1* mutant. Is it a loss-of-function mutant? Is a defective protein produced in soybean from this mutant allele? Is the protein inactive? The authors show that the protein produced from this allele in yeast can still interact with STF1/2. See also other comments below.

-Since soybean contains 10 SPA homologues with very high protein similarity, it is necessary to assess that the mutations introduced by CRISPR specifically affect only the structure of RIN1. For this, it would be necessary to sequence the corresponding region of all SPA homologues in the mutants.

-In the same sense, how confident are the authors about the specificity of the probe used for in situ hybridization? Are they only recognizing RIN1 or also other SPA homologues?

-In the interaction experiments (Y2H and BiFC), the respective negative controls are lacking. Empty vectors for Y2H. Fusion to a nuclear protein that does not interact with either RIN1 or STFs in BiFC. Also, since the authors observed interaction with both RIN1 and the protein produced from the mutant *rin1* allele (at least in Y2H), it would be interesting to evaluate if the "strength" of the interaction is similar (for example, using a more quantitative assay of reporter gene activation in Y2H; e.g. beta-galactosidase activity or growth from successive dilutions of yeast cultures).

-Connected to this, is it possible that the differential effect of the *rin1* mutation is not due to differences in the encoded protein but to differences in the amount of protein? It would be nice to know if the *rin1* mutation affects protein levels, but I understand that antibodies against RIN1 are not available. Did the authors check RIN1 transcript levels in RIN1 and *rin1* plants? Are they similar?

-The authors show that *rin1* plants have increased STF1/2 protein levels and ascribe this to lower degradation. It would be important to check STF1/2 transcript levels to rule out an effect on transcription. The same for the time dependent series, since the authors conclude that STF1/2 are degraded under dark and accumulated under light. Is it possible to perform this experiment (in soybean) in the presence of MG132?

-Line 238: "...this effect is weaker in the *stf1/stf2* background" Please, give the numbers and analyze if the difference is significant.

-About results shown in Fig. 4: Does *stf1/2* mutation abolish the increased expression of GA2ox7 observed in *rin1*? This would support the molecular mechanism proposed by the authors. It is noteworthy that the effect of STF overexpression on GA2ox7 expression seems to be considerably weaker than the effect of RIN1 mutation, which may indicate that RIN1 is also affecting another pathway.

-The results of Fig. S6 suggest that RIN1 may also have a role in determining flowering time. At this point, the reader wonders if plants with the *rin1* allele or the CRISPR mutants are affected in this parameter. These results are then presented under Discussion and shown in Fig. S11. Please, show the results of Fig. S11 after those of Fig. S6. Here, different effects on flowering time are observed for the CRISPR mutants and plants with the *rin1* allele. This is an indication that *rin1* is not simply a loss-of-function mutant.

-Fig. S1 and other Figs.: Please, mention in each legend under which condition plants were grown (field, greenhouse, long day, high density, etc.).

-Line 474: No Co-IP experiment is shown.

-Fig. 3 legend and probably other parts of the manuscript. Do not mention that STFs are degraded by RIN1. Rather, RIN1 induces or promotes STF1/2 degradation.

-Fig. S6: It is mentioned that t-tests were used for comparison but three different groups were compared. Please, explain. Also, the cultivars are listed in Table S5 (not S6 as specified).

Dear Reviewers,

Thank you all very much for your thoughtful and helpful comments on our manuscript that have assisted to strengthen the study.

According to your suggestions, we have revised the language carefully with the help of an individual proficient in scientific English. Please find below our revisions in response to the concerns raised by the reviewers. In addition, we have revised the abstract to meet journal requirements.

To Reviewer 1:

We appreciate your professional review and constructive advice. In response to your questions, we have addressed your concerns carefully. In addition, for the language issues you raised, and the points marked in the manuscript, we have improved the language carefully with the help of a native English expert.

Question 1: Lines 210-11, the sentence is not clear to me.

Response: Apology for our confusing description. Since we have re-performed the yeast two-hybrid (Y2H) experiment with the GAL4 system, we revised the sentences accordingly. The sentence Lines 210-11 in the text now reads “**To this end, we performed yeast two-hybrid analysis (Y2H), a pull-down assay and bimolecular fluorescence complementation (BiFC) to investigate protein-protein interactions between RIN1 and STF1 and STF2. RIN1^{Wm82} can interact with both STF1 and STF2 *in vivo* when expressed in yeast and tobacco, and *in vitro* (Fig. 3b-d). To determine whether *rin1* alters this interaction, we examined the interaction of RIN1^{HH43} and *rin1* with STF1 and STF2 in the GAL4 yeast two-hybrid system. *rin1* can still interact with STF1 and STF2, but the interaction strength is weakened (Fig. 3b)**”. And the new sentences are now listed from lines 229 to 236 in the manuscript.

Question 2: Lines 346-348, how many replicates? 20 plants in 2 m is a thin planting, normal is ~30 plants.

Response: Thank you for raising this very helpful technical advice. The description of the

planting method in this paragraph refers to the QTL detection and fine-mapping experiments. For F₂ phenotype evaluation, a total of 207 seeds were planted in a row with a 20 cm space between each seed, without replication. For F₃, 20 seeds derived from the corresponding F₂ generation were planted in each row and the average plant height and internode length of each row was considered as a repetition of F₂ generation. For every recombinant derived from F₆ progenies, 20 seeds were planted in each row, and phenotypes were recorded for fine mapping. Each time we grew the population, we also grew the parents HH43 and *rin1* as controls, following the same process with the population.

The *RINI* locus could be detected under normal planting densities, and the phenotypes of plant height and internode length could be observed at different densities, we thus used a thin density to observe the phenotype more conveniently. **We describe this part in line 414-436 of the manuscript.**

Question 3: Lines 354-55: mention what happened to row spacing and plant spacing within row with increase in planting density.

Response: Thank you for the recommendation. We have provided further details on the planting methods at different densities as **“Four replicates were performed in the natural field of Harbin in 2020. Plants were sown in middle of May, with three different plant densities (250,000, 350,000, and 450,000 plants per hectare). For 250,000 and 350,000 plants per hectare, seeds were only set in a row, and the average spacing between plants was 6.08 cm and 4.34 cm, respectively. For 450,000 plants per hectare, the row spacing remained at 60 cm. While the seeds were set in two rows with a zigzag shape, and the average spacing between plants was narrowed down to 3.38 cm. Plant height, number of nodes, internode length, grain yield per plant and grain yield per plot were recorded and calculated after harvest.”**. And this part is in line 437-445 of the manuscript.

To Reviewer 2:

Thank you very much for raising these very constructive recommendations, which are very helpful for the improvement of our article. In response to the language improvement, we

sought the professional help from native English specialist and improved substantially. The followings are our responses to your questions and advice.

Question 1: Since the authors link these parameters with grain yield, it would be interesting to know if these plants also show an increase in this parameter. In fact, in the last sentence of Results the authors conclude that “These results indicate that *rin1* is an ideal allele for improving soybean yield under dense planting conditions.” Is this specific for the *rin1* allele or knocking out *RIN1* as in the CRISPR mutants has a similar effect? This also leads to the question of the nature of the *rin1* mutant. Is it a loss-of-function mutant? Is a defective protein produced in soybean from this mutant allele? Is the protein inactive?

Response: Thank you advice these points which are very valuable to classify the functions of various alleles of *RIN1*.

RIN1/SPA3a protein contains three protein domains: a kinase domain, a coiled coil domain and a WD-repeat domain. The mutation of *rin1* results in the truncation of the WD-repeat domain but maintains the full domains of kinase domain and coiled coil domain. While both CRISPR-Cas9 mutants *rin1*^{CR1} and *rin1*^{CR2} are similar and deficient of all three domains (see below figure or Fig. 1e). We therefore speculate that *rin1* maintains partial protein functions while *rin1*^{CR1} and *rin1*^{CR2} proteins may be fully inactive. To this end, we performed the transient transformation assays in tobacco to verify the protein activity in regulating the expression of the promoters of *GA2ox7a* and *GA2ox7b*. The results showed that relative to the functional *RIN1*^{Wm82} and *RIN1*^{HH43}, the inhibitory effect of *rin1*^{CR1} on the promoter of *GA2ox7a* and *GA2ox7b* is completely lost, whereas the *rin1* can induce the partial expression of *GA2ox7a* and *GA2ox7b* (Fig. 4c,d). These results indicate that *rin1* may be a partial functional variant but the *rin1*^{CR1} and *rin1*^{CR2} proteins are the inactive ones. We therefore added these results in the manuscript from lines 284-291 as “**To further explore the functional activity among different alleles of *RIN1* to regulate the expression of *GA2ox7a* and *GA2ox7b*, we performed a dual-luciferase transient expression in tobacco leaves. The results demonstrated that relative to the functional *RIN1*^{Wm82} and *RIN1*^{HH43}, the inhibitory effect of *rin1*^{CR1} on the promoter of *GA2ox7a* and *GA2ox7b* is completely lost, whereas the *rin1* can induce the expression of *GA2ox7a* and *GA2ox7b*, although the ability is decreased**

(Fig. 4c-d). These results indicate that RIN1 can inhibit the expression levels of *GA2ox7a* and *GA2ox7b*, and *rin1* might be a partial functional variant.”.

Considering the high-yield phenotype of *rin1* with partial functional loss under dense planting conditions, we speculate that *rin1^{CR1}* and *rin1^{CR2}* also have the potential to increase yield at dense planting conditions since they are fully loss of functional variants. Due to lacking the seeds of *rin1^{CR1}* and *rin1^{CR2}* for field evaluation, we have not yet got the results, but these evaluations are in the process.

Fig.1 e

e Gene structures of the *rin1* candidate gene show three allelic variations in Wm82, HH43, HN35, *rin1*, *rin1^{CR1}* and *rin1^{CR2}*. +, Coding regions (CDS). AA, Amino acid.

Fig.4 c-d

c Constructs used for the transient transfection assay. *P35s*: 35s Promoter, *pGA2ox7a*: *GA2ox7a*

Promoter, *pGA2ox7b*: *GA2ox7b* Promoter. **d** *Luciferase (LUC)* activity under the control of the *GA2ox7a* or *GA2ox7b* promoter, the combinations in **(d)** link to the numbers in **(c)**. Data are means \pm SD from three dependent biological replicates. One-way ANOVA were performed to generate the *P* values, *P* < 0.05.

Question 2: Since soybean contains 10 SPA homologues with very high protein similarity, it is necessary to assess that the mutations introduced by CRISPR specifically affect only the structure of *RIN1*. For this, it would be necessary to sequence the corresponding region of all SPA homologues in the mutants.

Response: Thank you for raising this important point. In a view of the multiple copies of genes in soybeans, we have aligned the 10 soybean *SPA* genes to identify homologous segments where the *rin1* CRISPR–Cas9 targets were located and sequenced these segments in the background of the edited lines. The other nine *SPA* genes were not mutated in this region. The alignments and sequencing data are shown in **Supplementary Files 1**.

Question 3: In the same sense, how confident are the authors about the specificity of the probe used for *in situ* hybridization? Are they only recognizing *RIN1* or also other SPA homologues?

Response: Thanks for your kind suggestion. In the revised manuscript, we added the specific detection of the probe, and re-designed the specific probes for *RINI/SPA3a* and *SPA3b*, two closely duplicated paralogues for *in situ* hybridization at different growth periods of Wm82. The positions of the probes for *RINI/SPA3a* and *SPA3b* are showed below and **Supplementary Fig. 9b**. We then tested the probe specificity of both genes. The results showed that the expression of *RINI* could not be detected in *rin1^{CR1}* compared with wild-type Wm82, but the expression of *SPA3b* can be detected either in *RINI* or in *rin1^{CR1}* (**Supplementary Fig. 9c**). Thus, we confirmed that the probe we used in this article is specific to *RINI*. We therefore repeated the expression patterns of *RINI* use this newly designed probe at different developmental stages and replaced **Fig. 3a**. The results showed that *RINI* was expressed in shoot apical meristems, axillary meristems and leaf primordium. We also modified the descriptions in the manuscript as “ **We first confirmed the specificity of the *RINI* probe used for *in situ* hybridization. Compared with wild-type Wm82, *RINI***

expression nearly can not be detected in *rin1^{CR1}*, but the expression of *SPA3b* (which has the highest homology with *RIN1*) was not obviously affected either *RIN1* or *rin1^{CR1}* lines (Supplementary Fig. 9b,c). Next, we performed *in situ* hybridizations in shoot apical meristems isolated from Wm82 at 10, 20 and 30 DAE. *RIN1* is expressed in the shoot apical meristem, leaf primordia and axillary meristems (Fig. 3a).” in line 204 to 210.

Supplementary Fig. 9

b Schematic representation of the positions of the *RIN1* and *SPA3b* probes used for *in situ* hybridization in (c). **c** *In situ* hybridization to determine the spatial expression pattern of *RIN1* and *SPA3b* in the stem tips of Wm82 and *rin1^{CR1}*. Arrows represent the expression positions of *RIN1* and *SPA3b*. Scale bars = 100 μ m.

Fig.3

a Spatial expression of *RIN1* in Wm82 stem tips at different periods as assayed by RNA in situ hybridization. Arrows point to shoot apical meristems (SAM) and axillary meristems (AM). L: leaf primordium. Scale bars = 100 μ m.

Question 4: In the interaction experiments (Y2H and BiFC), the respective negative controls are lacking. Empty vectors for Y2H. Fusion to a nuclear protein that does not interact with either *RIN1* or STFs in BiFC. Also, since the authors observed interaction with both *RIN1* and the protein produced from the mutant *rin1* allele (at least in Y2H), it would be interesting to evaluate if the “strength” of the interaction is similar (for example, using a more quantitative assay of reporter gene activation in Y2H; e.g. beta-galactosidase activity or growth from successive dilutions of yeast cultures).

Response: Our apologies for lacking the negative controls. We have re-performed the interaction experiments, including Y2H experiment and the BiFC experiment, supplemented with additional negative controls. These additional controls further support the observation that *RIN1* can interact with both STF1 and STF2 (**Fig. 3b–d**), at least when heterologously express. For BiFC, we used a nuclear localized protein PRR5a as the negative control which was unable to interact with *RIN1*, STF1 and STF2 (**Fig. 3d**).

To detect whether variation in *rin1* affects the interaction with STFs, we added the Y2H experiments in the GAL4 system. We found that both RIN1^{HH43} and *rin1* could interact with STF1 and STF2, but the interaction is significantly weakened for *rin1* (Fig. 3b). We also modified the descriptions in the manuscript as “**To determine whether *rin1* alters this interaction, we examined the interaction of RIN1^{HH43} and *rin1* with STF1 and STF2 in the GAL4 yeast two-hybrid system. *rin1* can still interact with STF1 and STF2, but the interaction strength is weakened compared with full functional RIN1 (Fig. 3b)**” in line 233 to 236.

Fig. 3b

b Verification of the physical interaction of RIN1^{Wm82} and STF1/STF2 in Y2H system. p53:T for positive control, empty pGAD vector for negative control.

Fig. 3d

d BiFC analysis of physical interactions between RIN1^{Wm82} and STF1 and STF2. YFP^C-tagged RIN1^{Wm82} together with YFP^N-tagged STF1 and STF2, respectively, were transiently expressed in tobacco leaves. PRR5a (Glyma.06G136600), PRR5-YFP^C and PRR5-YFP^N were used for the negative controls. YFP^C: C-terminal YFP; YFP^N: N-terminal YFP. Scale bar = 20 μm.

Question 5: Connected to this, is it possible that the differential effect of the *rin1* mutation is not due to differences in the encoded protein but to differences in the amount of protein? It would be nice to know if the *rin1* mutation affects protein levels, but I understand that antibodies against RIN1 are not available. Did the authors check *RIN1* transcript levels in *RIN1* and *rin1* plants? Are they similar?

Response: Thank you for the good point. We have checked *RIN1* expression in the NILs-*RIN1* by qRT-PCR and found that mutation of *rin1* did not change *RIN1* transcript levels, indicating that the *rin1* variant likely instead affects protein activity rather than its transcription (**Supplementary Fig. 9e**). We also modified the descriptions in the manuscript as

“Additionally, we evaluated *RIN1* expression in NIL-*RIN1*^{HH43} and NIL-*rin1* over a 24-h period. From ZT0 to ZT24, *RIN1* expression was unchanged between two NILs (Supplementary Fig. 9e), indicating the *rin1* mutation does not reduce its transcriptional abundance.” in line 214 to 217.

Supplementary Fig. 9e

e Relative expression levels of *RIN1* in NIL-*RIN1*^{HH43} and NIL-*rin1*. All plants were grown in growth chambers under long-day photoperiods (16 h light/8 h dark). RT-qPCR data are normalized to the reference gene *Tubulin* (*TUB*), and error bars indicate \pm SD of three independent replicates.

Question 6: The authors show that *rin1* plants have increased STF1/2 protein levels and ascribe this to lower degradation. It would be important to check STF1/2 transcript levels to rule out an effect on transcription. The same for the time dependent series, since the authors conclude that STF1/2 are degraded under dark and accumulated under light. Is it possible to perform this experiment (in soybean) in the presence of MG132?

Response: The advice is very helpful. We have examined the expression level of *STF1* and *STF2* in NILs-*RIN1* and found that variation in *rin1* has no changes on *STF1* and *STF2* transcript levels (Supplementary Fig. 11a–b). We also modified the descriptions in the manuscript as “**The addition of the proteasome inhibitor MG132 to the reaction decreased the protein degradation of STF1 and STF2 by RIN1 (Fig. 3e,f). To examine**

whether *rin1* causes a change in *STF* expression, we quantified *STF1* and *STF2* transcription in NIL-*RIN1*^{HH43} and NIL-*rin1* over a 24-h period. *rin1* mutations did not cause statistically significant changes on *STF1* and *STF2* transcriptions (Supplementary Fig. 11a, b).” in line 244 to 249.

Supplementary Fig. 11a–b

a–b Relative expression of *STF1* (**a**) and *STF2* (**b**) in the SAM of NILs-*RIN1* plants. Samples were harvested from ZT0 to ZT24 at 10 DAE. Expression was normalized to the reference gene *Tubulin* (*TUB*), and error bars indicate \pm SD of three independent replicates. A Student’s *t*-test was used to determine statistically significant differences between control and treatment means. Plants were grown in growth chambers under long-day photoperiods (16 h light/8 h dark).

We have repeated the *in vivo* protein accumulation assay with the addition of MG132. We applied MG132 to NIL-*RIN1* seedlings at 8 DAE and harvested the shoot apical meristem samples two days later. The results suggested that *STF1* and *STF2* were less abundant in *RIN1*^{HH43} compared with *rin1*. In addition of MG132, the abundance of *STF1* and *STF2* were enriched in *RIN1*^{HH43} (**Fig. 3e**). These results indicate *RIN1* affects the protein abundances of *STF1* and *STF2*. We had revised our manuscript accordingly “***STF1* and *STF2* are less abundant in darkness but accumulate in the light. Meanwhile, *RIN1*^{HH43} promotes the degradation of *STF1* and *STF2* both in the light and in darkness (Fig. 3e, Supplementary Fig. 10a).**” in line 238-241 and “**The addition of the proteasome inhibitor MG132 to the reaction decreased the protein degradation of *STF1* and *STF2* by *RIN1***

(Fig. 3e,f).” in line 244-246.

Fig.3e

e Western Immunoblots indicating RIN1^{HH43} promotes the degradation of GmSTF1/STF2 *in vivo*, STF1/STF2 is more stabilized in protein extracts from NIL-*rin1* plants than NIL-RIN1^{HH43} plants growing in LD conditions (16h light/ 8h dark). ZT, Zeitgeber time; The membrane was probed by the anti-STF1/STF2 antibody; Anti-actin was used as a sample loading control.

Question 7: Line 238: “...this effect is weaker in the *stf1/stf2* background” Please, give the numbers and analyze if the difference is significant.

Response: Thank you for the kindness advice. We have now calculated the proportional decrease in internode length after introducing the *rin1* mutation into the *stf1 stf2* double mutant and *STF1/STF2* background, respectively, and annotated it in **Supplementary Fig. 13b**. In the *stf1 stf2* double-mutant background, the length of the first and second internodes was decreased by 8.7% and 23.6%, respectively. In the STF1/STF2 background, the reduction has risen sharply, the internode length of first and second nodes were decreased by 21.5% and 33.7%, respectively. The reduction rate of internode length in the two backgrounds is statistically significantly different (**Supplementary Fig. 13c**). We had revised our manuscript for this part as “**We also developed a NIL set for RIN1 and STF1 and STF2 in the same genetic background. The phenotypic effect of rin1 significantly weakened in the stf1 stf2 mutant background (Supplementary Fig. 13a-c), implying that RIN1 functions might be upstream of STF1 and STF2 and partially depends on STF1 and STF2.**” in line 266-270.

Supplementary Fig. 13a-c

a Phenotypes of *RIN1^{HH43}/STF1/STF2*, *rin1/STF1/STF2*, *RIN1^{HH43}/stf1/stf2*, and *rin1/stf1/stf2* under long-day photoperiods (16 h light/8 h dark). Scale bar = 5 cm. **b** Internode length of *RIN1^{HH43}/STF1/STF2*, *rin1/STF1/STF2*, *RIN1^{HH43}/stf1/stf2*, and *rin1/stf1/stf2* under long-day photoperiods (16 h light/8 h dark). All data are means ± SEM (n = 6 plants). One-way ANOVA was performed to determine statistical significance for the effect of genotype on the trait of interest, $P < 0.05$. **c** Significant differences analysis of the rate of plant height reduction in *rin1* under *STF1/STF2* and *stf1/stf2* backgrounds, respectively, in (b). A Student's *t*-test was used to determine statistically significant differences between control and treatment means.

Question 8: About results shown in Fig. 4: Does *stf1/2* mutation abolish the increased expression of *GA2ox7* observed in *rin1*? This would support the molecular mechanism proposed by the authors. It is noteworthy that the effect of *STF* overexpression on *GA2ox7* expression seems to be considerably weaker than the effect of *RIN1* mutation, which may indicate that *RIN1* is also affecting another pathway.

Response: Thank you for this valuable suggestion. According to your suggestion, we detected the expression levels of *GA2ox7a* and *GA2ox7b* in NILs of *RIN1^{HH43}/STF1/STF2*, *rin1/STF1/STF2*, *RIN1^{HH43}/stf1/stf2*, and *rin1/stf1/stf2*. The result shows that although *RIN1* can suppress the expression levels of *GA2ox7a* and *GA2ox7b* in both *stf1/stf2* and *STF1/STF2* background, the expression levels of *GA2ox7a* and *GA2ox7b* were significantly lower in the *stf1/stf2* background than in the *STF1/STF2* background (**Supplementary Fig. 13 d,e**), further confirming that *RIN1* is partially dependent on *STF1/STF2* which is in agreement with the

phenotypic observations of genetic test (Supplementary Fig. 13a-c). We had revised our manuscript for this part as “Since *RIN1* inhibits *GA2ox7a* and *GA2ox7b* expression while *STF1* and *STF2* promote its expression, we further examined the expression levels of *GA2ox7a* and *GA2ox7b* in *RIN1* NILs on the background of *STF1/STF2* and *stf1/stf2*, respectively. The result shows that although *RIN1* can suppress the expression levels of *GA2ox7a* and *GA2ox7b* in both *stf1/stf2* and *STF1/STF2* backgrounds, the expression levels of *GA2ox7a* and *GA2ox7b* were significantly lower in the *stf1/stf2* background than in the *STF1/STF2* background (Supplementary Fig. 13d,e), further confirming that *RIN1* is partially dependent on *STF1* and *STF2*.” in line 297-305.

Supplementary Fig. 13: *RIN1* is partially dependent on *STF1* and *STF2* for internode-length regulation.

a Phenotypes of *RIN1*^{HH43}/*STF1/STF2*, *rin1/STF1/STF2*, *RIN1*^{HH43}/*stf1/stf2*, and *rin1/stf1/stf2* under long-day photoperiods (16 h light/8 h dark). Scale bar = 5 cm. **b** Internode length of *RIN1*^{HH43}/*STF1/STF2*, *rin1/STF1/STF2*, *RIN1*^{HH43}/*stf1/stf2*, and *rin1/stf1/stf2* under long-day photoperiods (16 h light/8 h dark). All data are means \pm SEM (n = 6 plants). One-way ANOVA was performed to determine statistical significance for the effect of genotype on the trait of interest, $P < 0.05$. **c** Significant differences analysis of the rate of plant height reduction in *rin1* under *STF1/STF2* and *stf1/stf2* backgrounds, respectively, in **(b)**. A Student's *t*-test was used to determine statistically significant differences between control and treatment means. **d–e** Relative expression of *GA2ox7a* (**d**) and *GA2ox7b* (**e**) in the SAM of *RIN1*^{HH43}/*STF1/STF2*, *rin1/STF1/STF2*, *RIN1*^{HH43}/*stf1/stf2*, and *rin1/stf1/stf2* under long-day photoperiods (16 h light/8 h dark). Samples were harvested at ZT0 of 10 DAE. Expression was normalized to the reference gene *Tubulin* (*TUB*), and error bars indicate \pm SD of three independent replicates. Plants were grown in growth chambers under long-day photoperiods (16 h light/8 h dark). One-way ANOVA was performed to determine statistically significant differences.

Question 9: The results of Fig. S6 suggest that *RIN1* may also have a role in determining flowering time. At this point, the reader wonders if plants with the *rin1* allele or the CRISPR mutants are affected in this parameter. These results are then presented under Discussion and shown in Fig. S11. Please, show the results of Fig. S11 after those of Fig. S6. Here, different effects on flowering time are observed for the CRISPR mutants and plants with the *rin1* allele. This is an indication that *rin1* is not simply a loss-of-function mutant.

Response: Thanks for your advice. You are right. We have adjusted the presentation: the original Supplementary Fig. 6 is now to **Supplementary Fig. 7**, and the original Supplementary Fig. 11 is now **Supplementary Fig. 8**.

As for the question that *rin1* does not affect flowering time but *rin1*^{CR} affects the flowering time in soybean, we speculate the reason for two aspects. Firstly, the transient transformation result showed that *rin1* is a mutant with partial loss of function, but *rin1*^{CR} is full loss of function mutant. They are different from two protein domains of kinase or coiled coil which may have functions to control flowering. The other explanation is that this may be due to the differences between the background of these mutants which *rin1* is from the cross between

HH43 and HN35, while *rin1^{CR}* is from Wm82. The different backgrounds of *rin1* and *rin1^{CR}* may lead to flowering differences. The flowering difference between *rin1* and *rin1^{CR}* is very interesting and is under further investigation.

Question 10: Fig. S1 and other Figs.: Please, mention in each legend under which condition plants were grown (field, greenhouse, long day, high density, etc.)

Response: Thanks for these points. We have now described the growth conditions in each legend and described them in further detail in the **Methods**.

Question 11: Line 474: No Co-IP experiment is shown.

Response: Very sorry for our mistake. We now have changed the sentence in line 474 to “***In vivo degradation assays***” in line 584.

Question 12: Fig. 3 legend and probably other parts of the manuscript. Do not mention that STFs are degraded by RIN1. Rather, RIN1 induces or promotes STF1/2 degradation.

Response: Many thanks for your comments and fully agreed with this. We have changed all the phrasing throughout the manuscript to align with this.

Question 13: Fig. S6: It is mentioned that t-tests were used for comparison but three different groups were compared. Please, explain. Also, the cultivars are listed in Table S5 (not S6 as specified).

Response: Deep apology for these mistakes. We used the One-way ANOVA here for significant difference analysis. We corrected the significant difference analysis of

Supplementary Fig. 6 (Supplementary Fig. 7 in the previous reviewed manuscript) using One-way ANOVA.

The list of cultivars has now been corrected to **Supplementary Table 5**.

Reviewer #1 (Remarks to the Author):

My concerns from the initial review have been addressed in the revised manuscript. No further comment. This is paper with potentially impactful finding to increase soybean yield under high planting densities in highly fertile land.

I recommend publication of the revised manuscript without further revision.

Reviewer #2 (Remarks to the Author):

The authors satisfactorily addressed the questions raised in the previous round of review.

Just two remarks;

-Lines 292-295: The sentence starting "These results..." is duplicated.

-Line 808: I guess it's "independent" instead of "dependent"

Dear Reviewers,

Thanks again for your constructive suggestions to our manuscript.

To Reviewer 1:

Thank you for your recognition of our manuscript. Your professional suggestions are very helpful to improve our article.

To Reviewer 2:

We appreciate it for the thoughtful comments and suggestions on our manuscript.

According to the suggestions for this reviewed, we responded as follows:

-Lines 292-295: The sentence starting "These results..." is duplicated.

Response: We are so sorry for this mistake. We have deleted the superfluous sentences. The sentence is in lines 294-295 in the reviewed manuscript.

-Line 808: I guess it's "independent" instead of "dependent"

Response: Our apologies for the miswriting. We have changed "dependent" into "independent" in the legend of Fig.4. This part is in line 875 in the reviewed manuscript.